# Songbirds work around computational complexity by learning song vocabulary independently of sequence

Dina Lipkind[1], Anja T. Zai[2,3], Alexander Hanuschkin[2,3], Gary F. Marcus[4,5], Ofer Tchernichovski[1] & Richard H.R. Hahnloser[2,3]

While acquiring motor skills, animals transform their plastic motor sequences to match desired targets. However, because both the structure and temporal position of individual gestures are adjustable, the number of possible motor transformations increases exponentially with sequence length. Identifying the optimal transformation towards a given target is therefore a computationally intractable problem. Here we show an evolutionary workaround for reducing the computational complexity of song learning in zebra finches. We prompt juveniles to modify syllable phonology and sequence in a learned song to match a newly introduced target song. Surprisingly, juveniles match each syllable to the most spectrally similar sound in the target, regardless of its temporal position, resulting in unnecessary sequence errors, that they later try to correct. Thus, zebra finches prioritize efficient learning of syllable vocabulary, at the cost of inefficient syntax learning. This strategy provides a non-optimal but computationally manageable solution to the task of vocal sequence learning.

[1] Department of Psychology, Hunter College, City University of New York, New York, NY 10065, USA. [2] Institute of Neuroinformatics, University of Zurich/ETH Zurich, Zurich 8057, Switzerland. [3] Neuroscience Center Zurich (ZNZ), Zurich 8057, Switzerland. [4] Department of Psychology, New York University, New York, NY 10003, USA. [5] Geometric Intelligence, New York, NY 10013, USA. Correspondence and requests for materials should be addressed to D.L. (email: dina.lipkind@gmail.com) or to R.H.R.H. (email: rich@ini.ethz.ch)

Minimizing a cost or an error function is at the core of many biological and artificial learning mechanisms. Error minimization, or function optimization in the broader sense, underlies algorithms such as linear regression for minimizing residuals to linear fits, and the simplex algorithm for solving linear optimization problems[1]. Optimization strategies have been also elucidated in many adaptive behaviors[2], particularly in cases involving simple behavioral targets, such as the minimization of retinal slip during smooth[3] and ballistic[4] eye movements, or minimization of movement time[5], endpoint error[6], and movement variability[7] during arm reaching movements. However, many animals are also capable of learning complex behavioral sequences, such as courtship songs, that precisely match acquired sensory target sequences[8, 9]. What kind of optimization strategy can guide behavior toward such complex targets?

Natural learning of complex behaviors often requires adapting both the structure and the order of gestures in a sequence (Fig. 1a), which is a more complex task than adapting either gestures or sequence alone[10–14]. Consider the example of learning a complex word in a foreign language: if one's utterance is misunderstood, is it because some speech sounds (e.g., phonemes) were pronounced incorrectly (a structural error), or because they were performed in the wrong order (a timing error), or may be some combination of both? Finding an optimal way to reduce both structural and temporal performance errors constitutes a quadratic assignment problem (Supplementary Notes). Such optimization problems are computationally intractable, meaning there is no known efficient algorithm for solving them[15–17].

Songbirds, being skilled vocal learners[18–21], provide an opportunity for studying how errors are assigned and minimized during the learning of complex motor sequences. A young zebra finch (*Taeniopygia guttata*) imitating an adult tutor has to match a series of spectrally distinct sounds (syllables) performed in a precise order (Fig. 1b). Zebra finches are capable of adjusting their developing song towards its target in a variety of ways, including morphing the spectral (phonological) structure of song syllables[22–25], generating and adding novel syllables to their song[23, 25, 26], and rearranging the positions of existing syllables[26, 27]. How then do they cope with the complexity of selecting the appropriate combination of operations that would reduce the mismatch between their own song and the target?

A possible way to reduce computational complexity could be to optimize one aspect of the task, while ignoring the costs of the other. At one extreme, the task could be reduced to assigning each syllable in the bird's song to the temporally corresponding syllable in the target song (Fig. 1c, left). Such strategy would minimize sequence rearrangements, at the cost of possibly large phonological adjustments. Although this hypothesis has not been directly tested, a number of previous findings suggest that songbirds may not be using global alignment between song and target as a learning strategy. These include the observation that individual syllables are recognizable in developing zebra finch song before the correct sequence is apparent[28]; the existence of an early developmental phase in which repetitions of a single "proto-syllable" differentiate towards multiple targets[22, 24, 25, 29, 30]; the fact

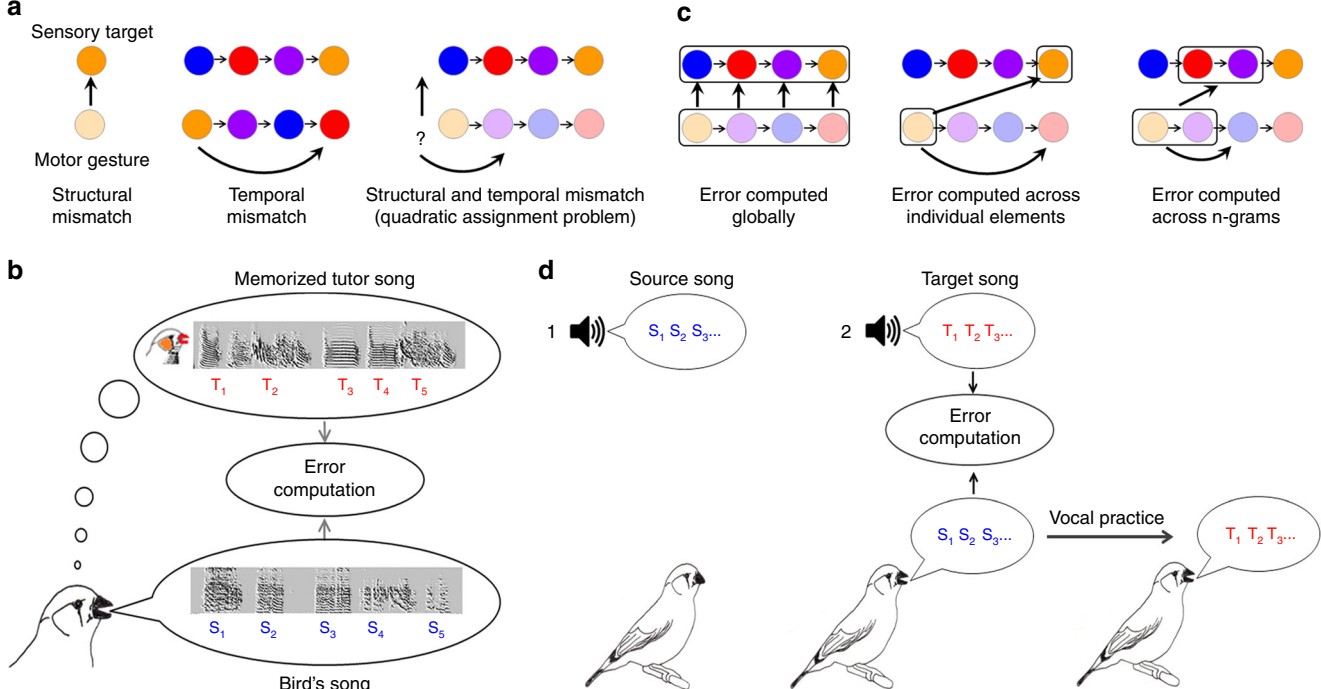

**Fig. 1** Motor sequence learning: hypotheses and testing method. **a** Sensorimotor learning may require adapting the structure of a motor gesture to a desired target (left, different colors indicate a structural mismatch); adapting the temporal order of gestures to a target sequence (middle); or adapting a sequence of unformed gestures to a target sequence (right). In the latter case, the number of possible combinations of structural and temporal adjustments (vertical and horizontal arrows) increases exponentially with sequence length. **b** Song learning in zebra finches: a juvenile male gradually matches its own unformed vocalizations (bottom sonogram) to a memorized song of a tutor (top sonogram). Letters represent consecutive syllables of the bird's song ($S_1$, $S_2$...) and the target ($T_1$, $T_2$...) **c** Hypothetical strategies of motor sequence learning: left, motor gestures are matched to temporally corresponding target gestures; middle, gestures are matched to the most structurally similar targets, minimizing structural changes, but possibly requiring considerable sequential rearrangements; right, error is computed across 'chunks' of gestures attempting to achieve local (non-optimal) tradeoffs between structural and temporal adjustments. **d** Experimental setup for testing vocal learning strategies. Left, a young bird hears a playback of an artificial song (source); once he learns it (middle), training is switched to a playback of a second song (target). The mismatch between the two songs is designed by the experimenter. The bird's vocal practice trajectory as it corrects the mismatch is continuously recorded

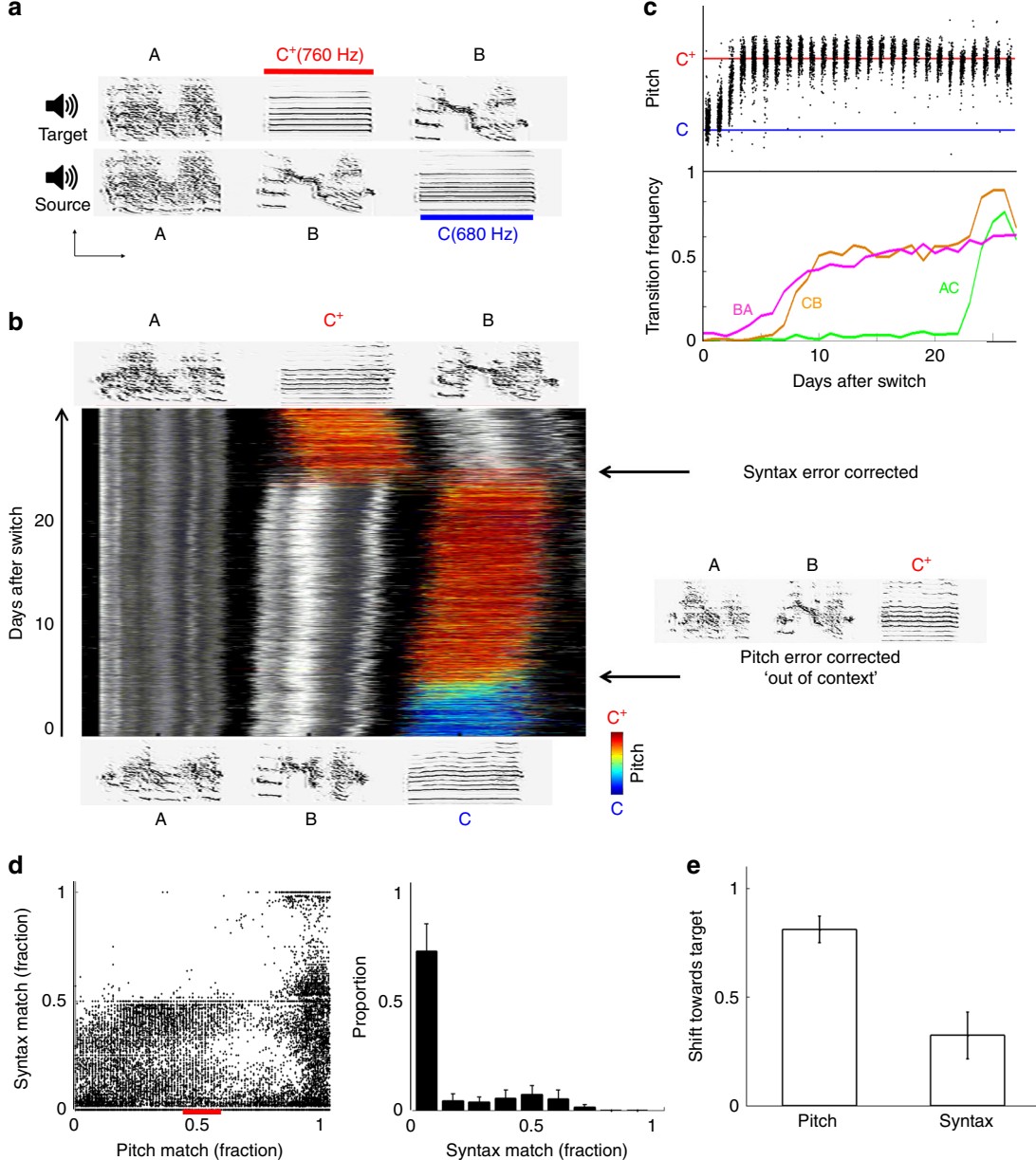

**Fig. 2** Phonological error correction in individual syllables disregards global similarity. **a** Song models used for imitation task 1 ABC → AC⁺B (a single motif is shown; birds were trained with two motif repetitions). Scale bars for sonograms are 100 ms (x axis) and 2 kHz (y axis). The pitch of syllable C⁺ in the target song is shifted up by two semitones with respect to C in the source. **b** Developmental singing trajectory of an experimental bird trained with imitation task 1. Stack plot shows consecutive renditions of song motifs containing C/C⁺ syllables, over experimental days (day 0, switch to target training; instances of the transition BA, which the bird acquired early on, see **c**, are excluded from this plot). Colors, pitch of C/C⁺; grayscale, Wiener entropy in neighboring syllables. Example sonograms of the bird's song at experiment start (bottom) and end (top). The bird first changed its song to ABC⁺ (sonogram on the *right*) and only *afterwards* corrected the syntax to AC⁺B (arrows). **c** Top, the median pitch of consecutive renditions of syllable C/C⁺ in the same experimental bird; bottom, the daily frequencies of target syllable transitions. **d** Left, scatter plot of the fraction of syntax correction vs. the fraction of pitch correction (0, source pitch/syntax; 1, target pitch/syntax) in consecutive data bins (bin size 30 samples with 25 samples overlap) across experimental birds trained with imitation task 1 (ABC → AC⁺B; n = 3) and imitation task 2 (ABC → A⁺C⁺B, only syllable C/C⁺ pitch correction included; n = 9; same for **e**); right, distribution of the fraction of syntax correction at 45–55% pitch correction (red horizontal bar on bottom of left panel); means ± s.e.m. across birds for each bin. When pitch reached half way to target, syntax was mostly unchanged. **e** Fraction of pitch and syntax correction at developmental endpoint (0, source pitch/syntax; 1, target pitch/syntax; means±s.e.m.)

that many songbird species perform variable syllable sequences as adults (e.g., nightingales, starlings and Bengalese finches); and the ability of zebra finches to match a target exclusively through syllable rearrangements, without changing phonology[26]. An alternative strategy, therefore, could be to assign song syllables to target syllables in a manner that minimizes phonological distances, while ignoring combinatorial distances (Fig. 1c, middle). Such phonological greediness would increase the number of ensuing sequence changes and thus the overall sequencing cost[26]. An intermediate strategy could be to seek a trade-off between minimizing structural and temporal errors, for example by independently matching parts of the song sequence (such as phonology in bigrams or trigrams[27]) to parts of the target sequence (Fig. 1c, right).

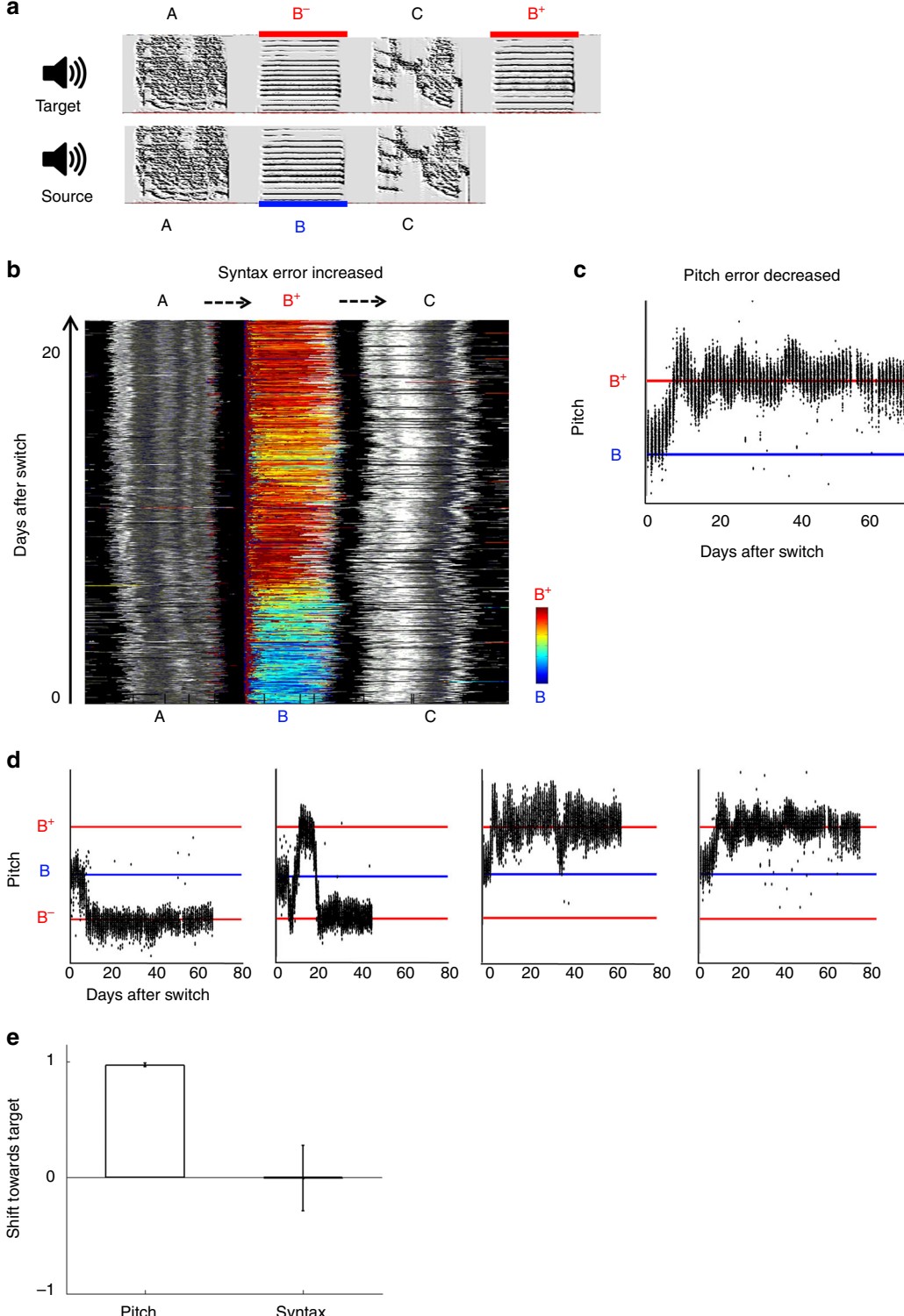

**Fig. 3** Phonological error correction is independent of temporal context. **a** In imitation task 3 (ABC → AB⁻CB⁺) syllable B in the source song is replaced by two similar, equidistant syllables in the target song: B⁻ in the same sequential context and B⁺ in a different context. **b** Stack plot of song motif ABC renditions in one experimental bird (colors as in Fig. 2b). Syllable B shifted to become target syllable B⁺, resulting in a decrease in syntax match due to acquisition of incorrect syllable transitions AB⁺ and B⁺C (dashed arrows). **c** The median pitch per syllable for all B/B⁺ renditions in the same experimental bird, showing accurate matching of the target syllable B⁺. **d** Pitch trajectories of experimental birds trained with the task (n = 4). Two birds shifted pitch down to the in-context target B⁻ (in one case after oscillations), and two others shifted pitch up to the out-of-context target B⁺. **e** Pitch and syntax shift towards target at developmental endpoint (0, source pitch/syntax; 1, full match of target pitch B⁻ or B⁺/full match of target syntax (first 3 syllables), defined as 100% performance of correct transitions; −1, syntax maximally shifted away from target, defined as 100% performance of incorrect transitions; means ± s.e.m. across birds)

To test which strategy zebra finches employ to learn their song, we used artificial tutoring to generate temporal and spectral mismatches between a bird's song and its target[23, 26, 31, 32]. We utilized the fact that under certain conditions (presumably the loss of a tutor due to high predation rates[33]) zebra finches can learn their song in a piecewise manner[25, 26] from more than one tutor[34–36]: we trained young male zebra finches with playbacks of an artificially designed tutor song (Methods section), and once we could reliably identify copies of all tutored syllables in the singing performance, we switched the training to an altered synthetic song (Fig. 1d). We continuously recorded the birds' vocal output and tracked the developmental trajectory of individual syllables, to uncover the underlying assignment of performance error and the manner in which it is minimized.

## Results

### Performance errors can be computed out of temporal context.
To test whether zebra finches compute vocal errors in syllables with respect to temporally corresponding targets (Fig. 1c, left), we first presented three birds with a three-syllable song ABC (source song). Once the birds copied it, we presented a variant (target) song with permuted syllable order (temporal mismatch) and with a shift in the pitch of one syllable (local spectral mismatch):

$$\text{Imitation task 1} \qquad A\,B\,C \rightarrow A\,C^+B,$$

(plus sign ($^+$) indicates a pitch mismatch of 1 or 2 semitones between syllables C and $C^+$; Fig. 2a; Supplementary Table 1, Supplementary Audio 1).

All three birds shifted the pitch of syllable C to the target pitch $C^+$ before making any changes in song syntax. This resulted in a performance of a song they never heard (A B C → A B $C^+$; Fig. 2b, Supplementary Audio 2–4), in which the "correct" syllables were sung in the "wrong" order. It took the birds several additional days to permute syllable order (fully or partially) towards the target syntax A $C^+$ B, one transition at a time (Fig. 2c), incorporating each new transition into their song and performing it in combination with the existing (source) transitions[26]. We tested an additional nine birds with a slightly more complex task involving pitch mismatches in two different syllables.

$$\text{Imitation task 2} \qquad A\,B\,C \rightarrow A^+\,C^+B$$

(The size and direction of pitch mismatches were varied across experimental birds; Supplementary Fig. 1a, Supplementary Table 2, Supplementary Audio 5–7). Note that no single alignment between the source and target songs can accommodate the pitch mismatches in both syllables. Nevertheless, the birds successfully corrected both mismatches, with no significant differences in success rate or speed (Supplementary Fig. 1b–d; percent of pitch correction at endpoint was $83 \pm 7\%$ vs. $79 \pm 7\%$ for A/$A^+$ and C/$C^+$ syllables respectively, and the speed of correcting 50% of the pitch mismatch was $10.8 \pm 4.0$ vs. $18.0 \pm 8.2$ days; NS for both comparisons, Wilcoxon rank sum test). Pooling results across tasks 1 and 2 ($n = 12$ birds), we found that when pitch correction reached 50%, only $15 \pm 8\%$ of the syntax mismatch was corrected. Therefore, the majority of pitch errors were corrected while sequentially misaligned with the target (Fig. 2d). This result indicates that zebra finches are not constrained to assign errors to song syllables according to their sequential order (Fig. 1c, left), but instead are able to compute vocal errors between song and target syllables at non-corresponding temporal positions (Fig. 1c, middle or right).

Overall, the birds corrected $81 \pm 6\%$ of the pitch error, but only $32 \pm 12\%$ of the syntax error (Fig. 2e). The syntax adjustments were smaller than in a previous study[26], which presented zebra

finches with the "pure" syntax correction task ABC → ACB ($57 \pm 1\%$ syntax error correction, $n = 17$), suggesting that phonological adjustments are prioritized over syntactical adjustments.

### Syntax error cannot bias phonological error assignment.
Imitation tasks 1 and 2 show that zebra finches are able to correct local pitch errors out of context, namely, without globally aligning their own performance with the target song. We next tested whether, when given a reasonable choice, the birds would prefer to match pitch in a correct alignment, so as to increase both local and global similarity with the target (Fig. 1c, right):

$$\text{Imitation task 3} \qquad A\,B\,C \rightarrow AB^-\,C\,B^+$$

(Where minus and plus signs indicate two semitone pitch shifts, down and up; Fig. 3a, Supplementary Table 3, Supplementary Audio 8; $n = 4$). In this task, syllable B in the source song is offered two fairly similar competing targets ($B^-$ and $B^+$), one of which is at the same sequential position ($B^-$), and the other at a different position ($B^+$). If the birds average those potential targets, the error should be zero, and we should observe little (if any) vocal changes. However, if the birds use a winner-take-all strategy for target selection, then the target chosen for syllable B would greatly affect the resulting global match: $B^-$ would increase global similarity to the target (A $B^-$ C), while $B^+$ would decrease it (A $B^+$ C).

All four birds shifted the pitch of syllable B, approaching either $B^-$ or $B^+$, with a remarkable accuracy of $97 \pm 1\%$, defeating the hypothesis of error averaging and indicating a winner-take-all target selection strategy. The remaining target syllable (towards which syllable B did not shift) was matched as well in 3 of the 4 birds, by a precursor initially performed outside of the song motif (see more on this below and in Supplementary Fig. 3). In contrast to early developmental stages where renditions of a single syllable prototype often differentiate to match distinct targets[22, 24, 25], we did not observe cases of duplication and "splitting" of syllable B renditions to match both $B^-$ and $B^+$ (except for a possible unsuccessful attempt in one bird, see bird 4 in Supplementary Fig. 3) However, surprisingly, only two birds chose target $B^-$ for syllable B resulting in a decrease in syntax error, while the other two chose the target $B^+$, resulting in an increase of syntax error (Fig. 3b–d; $-0.6 \pm 56.0\%$ syntax shift with respect to start point across birds; Fig. 3e). These mixed results suggest that syntax error does not affect phonological error assignment, and therefore point to the possibility of a dedicated phonology-correction mechanism that is "deaf" to syntax (Fig. 1c, middle).

### Phonological error correction is spectrally greedy.
Assuming a syllable-learning module that has no access to sequence information, how would such a module "decide" on syllable-to-target assignments? Our results so far appear to suggest that performance errors are assigned randomly among equidistant targets. We next tested if the size of the (small) phonological error matters to the bird, that is, if target selection is optimized (and greedy) with respect to spectral distance. We trained seven birds with a task offering syllables a choice between two spectrally similar targets, but this time the targets were at different pitch distances (1 vs. 2 semitones). We assigned a correct sequential context to the more distant target, and an incorrect sequential context to the closer target:

$$\text{Imitation task 4} \qquad A\,B\,C\,B^{+1} \leftrightarrow AB^{+2}\,C\,B^{-1}$$

(Superscript values indicate pitch shift size in semitones with respect to syllable B). We used two variants of this task, in which the source and target models were interchanged (task 4.1 ($n = 2$)

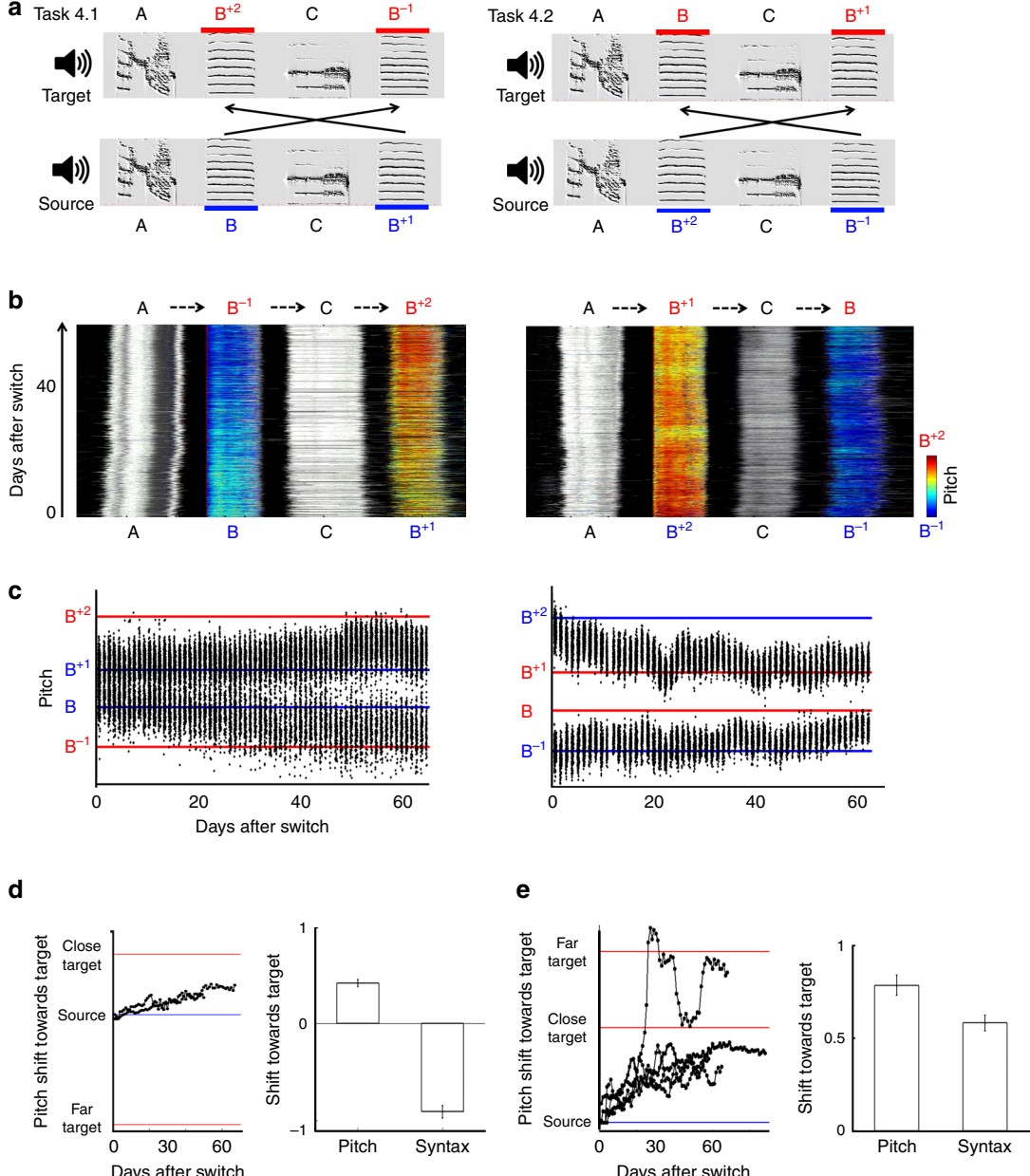

**Fig. 4** Phonologically greedy error correction. **a** Two imitation tasks (4.1, left; and 4.2, right) offering birds a choice to match syllables in the source song either to spectrally close targets (1 semitone) in the wrong sequential context (black arrows), or to slightly farther targets (2 semitones) in the correct context. **b** Developmental trajectories of two experimental birds trained with the imitation tasks in **a**. Stack plots show consecutive motif renditions following the switch to target training (day 0), colors indicate pitch of syllables $B/B^{-1}/B^{+1}/B^{+2}$; and grayscale indicates Wiener entropy in neighboring syllables (as in Figs. 2b and 3b). Birds shifted pitch towards the spectrally closer targets, resulting in a decreased (left) or an incomplete (right) syntax match due to the acquisition of two incorrect syllable transitions (dashed arrows). **c** Pitch trajectories of the same experimental birds, showing pitch shifts in the direction of the spectrally closer targets (see also Supplementary Fig. 2). Note that in task 4.1 (left), the closer and farther targets are in opposite pitch directions, while in task 4.2 (right) they are in the same direction. **d** Left, trajectories of pitch shift towards target across experimental birds trained with task 4.1 ($n = 2$; daily medians averaged over the two pitch shifted syllable types in each bird). Both birds shifted towards the close targets and away from the far targets. Right, pitch and syntax shift towards target at developmental endpoint (0, source pitch/syntax; 1, full match of either close or far target/full match of target syntax, defined as 100% performance of correct transitions; −1, syntax maximally shifted away from target, defined as 100% performance of incorrect transitions; means±s.e.m. across birds). Birds increased pitch similarity, but decreased syntax similarity to target. **e** Left, same as **d** left for birds trained with task 4.2 ($n = 5$). Four birds shifted pitch no further than the close targets, and one bird matched the far targets. Right, pitch and syntax shift towards target at developmental endpoint (0, source pitch/syntax; 1, full match of either close or far target/full match of target syntax, defined as 100% performance of correct transitions; means±s.e.m. across birds)

and task 4.2 ($n = 5$; Fig. 4a, Supplementary Table 4, Supplementary Audio 9).

Six out of seven birds shifted pitch towards the spectrally closer (1 semitone) targets (final pitch shift of $0.8 \pm 0.2$ semitones

toward target; Fig. 4b-e; Supplementary Fig. 2). Since the closer targets were at "incorrect" temporal positions, this choice resulted in a reduced match of the target syntax ($-91 \pm 6\%$ relative to source song, task 4.1, Fig. 4d), or in an incomplete match

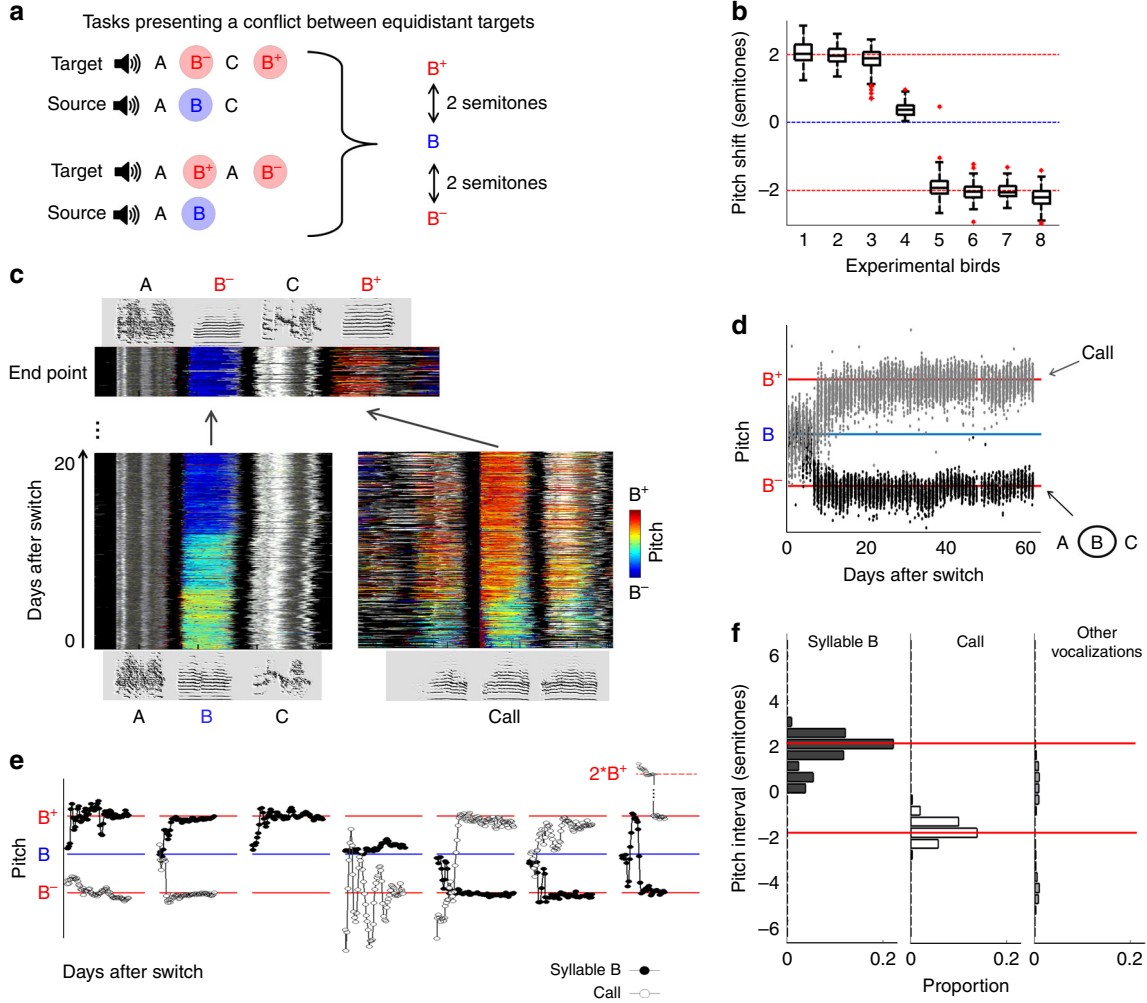

**Fig. 5** Competitive error assignment. **a** Imitation tasks presenting birds with a conflict between spectrally equidistant targets: ABC → AB⁻CB⁺ (imitation task 3, top) and AB → AB⁺AB⁻ (imitation task 5, bottom). Syllable B in the source songs is presented with the target syllables B⁻ and B⁺, shifted by 2 semitones down and up respectively. **b** Boxplots of the pitch shift from source of syllable B renditions on endpoint day, in experimental birds trained with the imitation tasks in **a** ($n = 4$, ABC → AB⁻CB⁺ $n = 4$, AB → AB⁺AB⁻). Birds are presented in descending order of median pitch shift. In seven out of eight birds, syllable B closely matched either B⁻ or B⁺. **c** Developmental trajectory of an experimental bird trained with ABC → AB⁻CB⁺. Stack plots as in Figs. 2–4. Syllable B in the song motif shifted down to match target syllable B⁻ (left); in parallel, a call performed outside the song motif differentiated into the missing target B⁺ (right), and got incorporated into the motif (top). **d** Median pitch in consecutive renditions of syllable B/B⁻ (black), and the call that shifted to B⁺ (gray), during development in the same bird. **e** Pitch trajectories (daily medians) of all other experimental birds trained with the imitation tasks in **a**. Black circles, motif syllable B; white circles, a vocalization type (typically a call), which shifted towards the vacant target. **f**, Left, distribution of the absolute pitch distance from source of renditions originating in motif syllable B on endpoint day (0 = pitch of source syllable B; mean per bin across experimental birds, $n = 8$); middle and right, same for harmonic vocalizations besides motif syllable B, within 6 semitone distance from source; for each bird, pitch distance is normalized by the direction to which syllable B has shifted: positive values for renditions shifted in the same direction as syllable B, and negative values for renditions shifted in the opposite direction. Across birds, no vocalization converged on the target occupied by the motif syllable. A call converged on the vacant target (middle), while all other vocalizations converged on neither of the targets (right)

(58 ± 7%, task 4.2, Fig. 4e). One-semitone targets were chosen regardless of their position in the target motif (2ⁿᵈ or 4ᵗʰ syllable in the motif, Fig. 4b–c; Supplementary Fig. 2), indicating that the choice was not a result of a salience effect of the last syllable in the motif. Thus, although we failed to bias the pitch trajectories of song syllables by offering the birds a greater sequence match, even small differences in local spectral distance affected the choice of targets for phonological error correction. These results confirm the conclusion that zebra finches employ a dedicated mechanism for learning the syllable vocabulary of their target song. Further, this mechanism is spectrally greedy, namely, it prioritizes local phonological match over global syntax match.

**Competitive syllable-target assignments**. Is phonological greediness sufficient to successfully learn a syllable vocabulary? What happens if there are unmatched syllables in the target song even after all syllables in the bird's song have converged on a target? While in early development multiple targets are often matched by duplication and differentiation of a single precursor syllable[22, 24, 25], we know that during later vocal development new syllable types can be added to the song to match missing targets[23, 25, 26]. Given our evidence for phonologically greedy target selection, we wondered how a syllable is recruited to an unoccupied target, when other (occupied) targets may be spectrally closer. To find out, we returned to the results of imitation task 3 (ABC → AB⁻ CB⁺, see above), and combined them with an

additional task, in which two equidistant target syllables are competing for a single source syllable, but no sequential bias is introduced:

$$\text{Imitation task 5} \qquad A\,B \rightarrow AB^+\,AB^-$$

(Where plus and minus signs indicate two semitone shifts up and down; Fig. 5a, Supplementary Table 5, Supplementary Audio 10). As in imitation task 3, we found that birds trained with imitation task 5 selected and matched one of the two equidistant targets, $B^+$ or $B^-$ ($83 \pm 22\%$ pitch error corrected), seemingly at random, confirming a winner-take-all mechanism for target selection across tasks 3 and 5 (Fig. 5b).

We next examined if and how did the birds in both tasks match the remaining target ($B^+$ or $B^-$), which was not selected by syllable B. We examined each bird's singing repertoire at the end of development, searching for vocalizations with pitch in the vicinity of the vacant target and back-tracking their developmental origin[23, 37, 38]. We found that 7 out of 8 birds matched the vacant target (endpoint pitch error of $0.22 \pm 0.06$ semitones) by adding a new vocalization type to their song motifs, in most cases originating from a call (Fig. 5c–e; Supplementary Fig. 3). We then further searched the birds' entire vocal output (including calls performed outside of song bouts) for any additional precursors that might have converged on either of the targets $B^+$ or $B^-$. To do so, we focused on vocalizations in the spectral vicinity of targets $B^+$ and $B^-$ (i.e., harmonic sounds with pitch range of $B^{-6}$ to $B^{+6}$ semitones), excluding from the analysis vocalization types originating in syllable B or the call that differentiated to match the vacant target. We did not find any additional vocalization types that converged on either of the targets (Fig. 5f). Thus, matching efficiency was close to the theoretical optimum: 7 out of 8 birds managed to successfully match both targets $B^+$ and $B^-$, and achieved this feat using only two motor precursors: they recruited syllable B to match one of the targets, and a call to match the other. The likelihood of such efficient matching to occur via phonological greediness alone is very small: assuming randomly distributed call origins, a purely greedy mechanism would achieve such high efficiency (to match the vacant but not the occupied target) with very low probability ($p = 0.5^7 = 0.008$, $n = 7$ birds). Therefore, our findings demonstrate an additional competitive constraint that prevents more than one motor syllable to converge on a single target syllable. In combination with a winner-take-all target selection, these constraints can ensure successful one-to-one matching of a syllable vocabulary. Such set of constraints is analogous to a "musical chairs" game, where players (motor syllables) compete for the occupancy of chairs (target syllables), and which ends with all chairs being occupied by a single player each.

**Error computation is modular across levels of song hierarchy**. The complete disregard of temporal context in the computation of phonological errors that we find contrasts with the fact that vocal imitation in songbirds (and zebra finches in particular) is clearly sensitive to temporal order. Zebra finches can precisely imitate entire syllable sequences, whether trained naturally with live tutors[22, 39], artificially with a single song early in development[24, 37], or serially with two songs that are switched in mid development[23, 26]. Furthermore, they do so even when presented with a "pure" combinatorial error (namely, with source and target songs that are composed of phonologically identical syllables, but that differ in syllable order), and at late stages of their sensitive period (up to days 90–120 post hatch[26]). Consequently, the large syntax errors due to greedy matching of phonology, which we observed in our experimental birds (especially in tasks 3–4), are not likely to result from a general indifference to

syntax imitation or from an age-related decline in the tendency to imitate syntax.

We therefore next examined whether birds trained with tasks 3–4 made any attempts to match the target song syntax, despite their greedy and context-insensitive phonology matching strategy. We found that most birds adjusted their song syntax towards the target, by partially correcting sequence errors that arose from greedy pitch matching, and by incorporating the newly acquired syllable (in task 3) into the appropriate position in the song motif (a task requiring the acquisition of two new syllable transitions). Overall, across experimental birds (tasks 1–5), 77% (20/26) made syntax adjustments towards the target, though only 15% (4/26) achieved a full match of the target syntax (Supplementary Fig. 4). The degree of final syntax match was not significantly correlated with the age of switching to target training ($R = -0.13$; $p = 0.58$). These results are consistent with previous findings showing that song syntax learning is a slow and piecemeal process, in which individual syllable transitions are acquired one by one, often leading to incomplete imitation in zebra finches trained serially with two songs[26]. Because the syntax "deaf" learning strategy for phonology that we discovered is on its own insufficient for correct imitation of a syllable sequence, our findings, taken together, indicate that error computation during song learning is carried out by two distinct and independent modules: one for acquiring the target syllable vocabulary and another for learning the target syntax.

## Discussion

We find that young zebra finches use a remarkably simple strategy to learn complex vocal sequences: they greedily match their syllable vocabulary to the target vocabulary, and initially ignore ensuing problems of temporal ordering, which are later resolved via an independent, slower process. A similar division of a complex learning task across distinct modules, sometimes with different time scales, was demonstrated in human sensorimotor learning[40, 41], and decision making[42, 43]. It is also akin to artificial learning algorithms in which texts are divided into separate bag-of-words (vocabulary) and n-gram (syntax) models[44]. Thus, zebra finches break down the computationally difficult task of exploring the entire space of possible motor permutations, into two simpler tasks, yielding a search for solutions that is non-optimal[45–47], but manageable.

Though zebra finches do not possess a globally optimal song learning strategy, our findings show that their strategy for learning a syllable vocabulary is very efficient, and even close to optimal. The complex developmental trajectories of song syllables which we observed could be accounted for by two computational "rules": (1) greedy (context-independent) matching of the phonological structure of target syllables; and (2) competition among syllables and targets over syllable-target associations. This process can be accurately described as a "musical chairs" game, beginning with a group of players (motor syllables) and a row of empty chairs (sensory targets), and ending with each chair being occupied by one player. Computationally, zebra finches' strategy for learning a syllable vocabulary amounts to solving a linear assignment problem[48] (Supplementary Notes), a fundamental optimization problem consisting of minimizing the cost of assigning a group of agents to perform a group of tasks (for example, assigning taxi cabs to passengers so as to minimize overall pick-up time). Interestingly, the same strategy is employed in the currently most successful method for automatic evaluation of document similarity, known as word mover's distance[49, 50] (Supplementary Notes).

Engineers use efficient methods to solve linear assignment problems[51]. However, it is not known how such problems are solved by biological systems, in which agent-task assignments

must be estimated from noisy input, such as the variable song performances of juvenile songbirds and their tutors. Therefore, we find that zebra finches' explorative and greedy strategy is more closely paralleled by machine learning algorithms such as the expectation maximization (EM) algorithm[52], in which models are gradually matched to observed data in an iterative process. If we view each rendition of a target syllable as an observable data point and each of the birds' own syllables as a Gaussian model with unknown (hidden) parameters, then birds' greedy-competitive strategy of maximizing the overlap between model and data can be approximated by the EM algorithm for Gaussian mixture models (Supplementary Notes, Supplementary Fig. 5).

What sort of neural architecture could implement the greedy-competitive algorithm employed by zebra finches? Answering this question can help to restrict the space of possible neural models for sensorimotor vocal learning. In most models, the brain is assumed to compute a global error by aligning the motor output with the entire target sequence and performing temporal summation over the partial errors[21, 53–55]. Our findings require the modification of such models and more generally, of models assuming simple one-to-one links between neural representations of motor gestures and sensory targets. The "musical chairs" style competition we observed suggests a more complex dynamic network, where syllable-target associations are shaped according to linear assignment constraints, such as for example Hopfield and Tank's constraint-satisfying attractor network[56]. Our results are consistent with models in which song features are reinforced when they are sufficiently close to any target syllable[57], and where spectral learning and sequence learning are carried out independently by two separate modules[58–60].

What are the smallest units at which spectral learning is independent of temporal context? We find a phonologically greedy and context-independent target matching strategy at the time scale of syllable vocabulary, but what happens at time scales within individual syllables? Does context independence continue all the way to the level of the smallest controllable song segments (5–10 ms)[61], or does it break down at a certain point along the song hierarchy? The answer depends on which are the smallest units within the song that a bird is able to rearrange. Namely, units that cannot be "moved around" with respect to each other must be learned in the appropriate context, since in such a case, sequence errors resulting from greedy learning could not be corrected. Given the laborious and time-consuming nature of sequence rearrangements[26], learning individual vocal gestures in a context-independent manner would be a highly inefficient strategy, as it would require a large number of positional rearrangements. In addition, resolving competitive conflicts between spectrally similar targets at such small time scales might be difficult, because many more conflicts would need to be considered, thus substantially increasing the number of computations to be performed. A more reasonable strategy would be to switch from context "deaf" to context-dependent learning at small time scales. Previous evidence of in-situ differentiation of sub-syllabic elements during zebra finch development[24] suggests that the transition point from phonologically greedy to context-dependent target matching may be the song syllable. This would imply that zebra finches could not rearrange the positions of sub-syllabic elements within their song. However, large (50–100 ms long) sub-syllabic notes have been shown to constitute behavioral breaking points[62], and are thought to be carried out by distinct neural activation chains in the premotor song nucleus HVC[63]. Therefore an alternative hypothesis is that such sub-syllabic notes are rearrange-able, and that their spectral structure is learned in a phonologically greedy manner. These hypotheses could be tested in the future with appropriate serial tutoring tasks.

Our serial tutoring paradigm is confined to mid and late stages of song development, at which the early phenomenon of duplication and differentiation of syllable prototypes to match multiple targets[22, 24, 25, 29, 30] is no longer observed. In our experiments, when a syllable was offered two spectrally similar targets (tasks 3 and 5), only one of those targets was selected—the syllable never "split"[25] to match both. We therefore do not have direct evidence as to what target assignment algorithm may govern the duplication and differentiation of early proto-syllables, and even on whether proto-syllable splitting is related to target assignments at all or occurs prior to targets being assigned. We cannot rule out the possibility that the "musical chairs" constraints we discovered might not (or not fully) apply to early song development. If our model applies to earlier stages of song development, this would mean that duplicated proto-syllables differentiate towards targets chosen by a phonologically greedy and competitive algorithm. In such a case, adjacent renditions of a proto-syllable, would not necessarily differentiate towards temporally adjacent syllables in the target song, but would instead differentiate towards targets according to spectral similarity. Since the spectral structure of early proto-syllables varies considerably across renditions[23, 29, 37], an interesting possibility is that target assignments may be initially determined by random spectral variation among proto-syllable renditions, and later reinforced by competitive musical-chairs-like constraints.

A previous study reported variable learning strategies in young zebra finches, including a "motif-learning" strategy[22], where syllables in mature zebra finch song develop from very early precursors already arranged in the correct sequential order. This suggests that zebra finches are capable of employing a global alignment target matching strategy (Fig. 1c, left) at early stages of their development. Another study of early development observed rare cases of motif learning[25], and showed that they originated from an earlier stage of duplication and differentiation of a single proto-syllable precursor, which is consistent with a non-global phonologically greedy strategy (since phonologically greedy target matching could lead to correct sequence matching by chance in rare cases). The challenge in resolving the question of target assignment strategies during early development, and in particular of testing the predictions of our model for early syllable precursors, lies in being able to identify specific assignments between variable proto-syllable renditions and their targets as early as possible. A recent step towards achieving this goal[25] was recording the activity of neurons in the premotor song nucleus HVC during singing in young zebra finches, allowing tracking of the process by which a single proto-syllable splits into two or more neurally distinct syllable precursors. Interestingly, HVC premotor neurons have been shown to exhibit precise auditory responses to the target song in young songbirds[32, 64], possibly providing an instructive signal for song development[32]. Therefore, combining recordings from HVC during singing in juveniles with recordings of auditory responses to the target song in the same neurons may, in the future, allow tracking the process by which targets are assigned to early syllable prototypes.

We speculate that zebra finches' modular vocal learning strategy is an evolutionary compromise between the need to efficiently use motor plasticity resources, and the computational burden of searching through a large space of possible solutions for the most efficient one. We do not know whether the same strategy is employed by other species of vocal learners, or in non-vocal learning processes, but in principle, such trade-off problems are inherent to any case of learning complex motor sequences (e.g., ref. [65]), and therefore may pose a common constraint shaping the evolution of motor learning mechanisms. A structurally greedy workaround, such as we observe in zebra finches, may be particularly useful for the learning of complex behaviors

in which combinatorial flexibility is essential. For example, many vocal learners (including songbird species such as nightingales and starlings) have much greater degrees of vocal combinatorial complexity than zebra finches. In particular, combinatorial diversity is a crucial characteristic of human languages, all of which consist of very large word vocabularies, from which an infinite number of higher order sequences can be generated. Thus, human infants' learning targets are the variable utterances of adults, in which basic speech sounds appear in diverse sequential contexts. Even ignoring the semantic and grammatical aspects of language, the task that human infants face of learning such highly diverse vocal repertoires from scratch seems extremely challenging. A sequence-independent and phonologically greedy vocal learning module may enable infants to extract a small set of basic learning targets (phonemes or syllables) from the highly variable input of adult tutors, and thus facilitate the acquisition of the speech sound vocabulary of their language, presumably occurring at the vocal babbling stage of infant development[66].

## Methods

**Experimental design**. Animal care and experimental procedures were conducted in accordance with the guidelines of the US National Institutes of Health and have been reviewed and approved by the Institutional Animal Care and Use Committee of Hunter College.

Male zebra finches were bred at Hunter College, and reared in the absence of adult males between days 7–30 post hatch. Afterwards, birds were kept singly in sound attenuation chambers, and continuously recorded. From day 33–39 until day 43 birds were passively exposed to 20 playbacks per day of the source song, occurring at random with a probability of 0.005 per second. On day 43, each bird was trained to press a key to hear song playbacks, with a daily quota of 20. Once birds learned the source song, we switched to playbacks of the target song. Learning of the source was assessed by quantifying the percent of similarity (Sound Analysis Pro[38, 67]) between the bird's song motifs and the source model motif in 10 randomly chosen song bouts per day. We considered the source song as being learned when the similarity to the model was at least 70%. Since the sensitive period for song learning in zebra finches ends around day 90–100 post hatch, we had to select for relatively fast learners of the source song. Therefore, in tasks 1–3 and 5, we used only birds that learned the source before day 68 (mean switch day $62.0 \pm 0.8$; $n = 20$, 39% of the total birds trained with the source). Because the source song models in task 4 were more complex than in the other tasks, we extended the switch threshold to day 84 for this task (mean switch day $72.0 \pm 2.5$; $n = 7$; 12% of total birds trained with the source). Recording and training were done using Sound Analysis Pro[38, 67], and continued until birds reached adulthood (day 99–158 post hatch). At these ages, males are sexually mature, and perform a crystalized song motif, which remains unchanged for the remainder of their lives[33].

Source and target song models were synthetically composed of natural syllables. Each model included either one or two harmonic syllables, which we used to generate pitch mismatches between source and target syllables (GOLDWAVE v. 5.68, www.goldwave.com). Each playback of a model included two motif renditions. To control for model-specific effects, we varied baseline pitch, and pitch shift size and direction across experimental birds. Supplementary Tables 1–5 contain a description of the training models used for each experimental bird. Sound files of the models are in Supplementary Audio 1 and 5–10.

**Data analysis**. Song feature calculation and cluster analysis were performed using Sound Analysis Pro[38, 67], on randomly selected 10% of the sound files in each developmental day. The rest of the analysis was performed using Matlab (Mathworks Inc.). Cluster information was used to elucidate the order of syllable types sung, and to track changes in spectral structure of syllable types, in particular shifted pitch. To distinguish between spectrally close syllable types (e.g., syllables B and B$^{+1}$ in task 4, see main text), which were sometimes lumped into a single cluster, clustering was refined by visual inspection of Wiener entropy stack plots of song motifs aligned on renditions of those syllable types, and sorted according to preceding or following context.

The percent of clustered syllables in bouts (assessed by manual inspection of a sample of 10 song bouts per bird in 10 randomly selected experimental birds) was $95.7 \pm 1.2\%$ at start point (day on which the training was switched from source to target), and $96.7 \pm 1.0\%$ during the transition from source to target. To test for possible biases of our clustering method, all song syllables of 8 of the birds were also clustered using a nearest neighbor classifier trained on visually clustered 70 ms log-power sound spectrograms segments, yielding similar results.

For pitch calculation in every millisecond of sound we used the YIN algorithm[68], implemented in Sound Analysis Pro[38, 67]. To track developmental pitch shifts in syllable and call types, we used the median pitch in each syllable or call rendition. Initial noisy segments characteristic of distance calls[69] were not included in the median calculation. The fraction of pitch error corrected in a syllable type on a given day (or data bin) was estimated from the median across all renditions on that day.

Song bouts were defined as sequences of identified syllable types with inter-syllable stop durations of less than a maximum duration that was determined by the typical stop duration in the endpoint song (150–200 ms). To quantify syntax error correction with respect to a pitch shifted syllable type, we tracked the identity of the preceding (converging) and following (diverging) syllable in each song bout, and calculated the fraction of renditions belonging to the target syntax on a given day (or data bin). For example, in Task 1 (ABC → AC$^+$B, see Results section), syllable A is the converging target syntax with respect to syllable C/C$^+$, and syllable B is the diverging target syntax. The overall fraction of syntax error correction was the mean of the fraction of converging and diverging syntax error correction. For comparison of syntax error correction with data from a previous study[26], in which birds were trained with the purely combinatorial task ABC → ACB, we calculated the sum of the fractions of performing the three target bigrams (AC, CB, and BA).

**Code availability**. All MATLAB codes used for analysis are available from the authors upon request.

**Data availability**. All relevant data are available from the authors upon request.

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

## Acknowledgements

We thank L.C. Parra, R. Douglas, and T.J. Sejnowski for critical reading of the manuscript, and K.A. Katlowitz for useful discussions. We thank P. Indyk and A. Backurs for pointing out to us the similarities between birdsong learning and the minimum common string partition problem. This work was supported by the Swiss National Science Foundation (grant 31003A_127024) and the European Research Council under the European Community's Seventh Framework Programme (FP7/2007–2013 / ERC Grant AdG 268911) to R.H.R.H., and by US Public Health Service grant (DC04722–137) and National Science Foundation grant (1261872) to O.T.

## Author contributions

D.L., O.T. and R.H.R.H. designed the research; D.L. performed the experiments. D.L., A.T.Z. and A.H. analyzed the data. A.T.Z. and R.H.R.H. developed computational models. D.L., G.F.M., O.T. and R.H.R.H. wrote the paper.

## Additional information

**Competing interests:** The authors declare no competing financial interests.

