## [Peer Review File · Nature Communications]

Reviewers' comments:

Reviewer #1 (Remarks to the Author):

I previously reviewed this paper for a different journal. While very little has changed in this re-submission, I outline below why I think this is an extremely important contribution to the birdsong field and to sensorimotor learning more generally, and is appropriate for publication in Nature Communications. There are several issues that I raised previously that I think could be addressed in discussion without further experiments. As I specify below, the main issue is tailoring the conclusions to the specific experiments and better contextualizing the findings with past work, specifically on sub-syllable matching and on duplication/differentiation strategies for learning.

Lipkind et al identify algorithms that zebra finches implement as they learn to match motor sequences to sensory targets. This is an important question that has the potential to generalize to broad classes of sensorimotor learning. One outstanding question is how animals deal with the problem of simultaneously matching both the structure and sequencing of targets. To address this issue, Lipkind et al build on previous work where they expose birds to an initial tutor song and observe how birds modify their song after exposure to a second tutor. In this study, they first develop a theoretical framework that formalizes the computational challenges of phonological and sequence learning. To identify strategies that the bird employs to reduce these challenges, they specifically devise tasks that cause direct 'conflict' between syllable phonology and syllable sequence. In this two-tutor paradigm, many birds prioritize phonology matching at the expense of sequence matching, in violation of predictions of a global alignment hypothesis of error correction. The main strength of the paper is that the training sets are very creatively designed to specifically pit phonology versus sequence. For example, the design of imitation task 4 was particularly clever and showed very nicely that the birds can indeed be deaf to sequence (and acoustically greedy); they did not choose to move a pitch of the syllables a little more (presumably a small increase in cost), with the benefit of keeping the sequence right. The core observations are clearly presented and for the most part well analyzed and show that under certain conditions many zebra finches exhibit a clear tendency to prioritize phonology matching, even at the cost of sequence matching. A second strength is that the supplement provides a much-needed formalism for the problem of target-matching that will be useful for the birdsong field.

While the big concept - that global alignment is not always deployed - is generally consistent with past work, the present paper provides a more definitive and rigorous demonstration of the idea. One main issue is I would like to see addressed, at least in discussion, is how their findings can generalize to the sequencing of sub-syllable elements into syllable targets, to earlier developmental stages, and to different species that as adults have variable sequencing. I outline these concerns below.

1.

1.1. The main experimental finding was that the birds were able to move the pitch out of context, in direct conflict with a global alignment model of error correction. While the present paper demonstrates this more elegantly than past work, it was already suggested by previous work that birds can do out-of-context comparisons and that the global alignment hypothesis was not universal. For example Immelman (1969, not cited) found that syllables imitated from a model song appear early in development before the appropriate serial order of syllables was apparent, showing both that phonology is prioritized and that it can be learned independent of sequence. Liu et al (2004, cited) showed that zebra finches (including brothers) use a variety of strategies to learn the same tutor song. Early in development (<55 dph) many birds serially repeat a single syllable; this single syllable subsequently duplicates and independently differentiates into multiple 'descendant' syllables. This same duplication and differentiation strategy has been demonstrated in several papers:

Tchernichovski et al (2001, e.g. Fig 5), in the 'protosyllables' described by Aronov et al. (2011, not cited), and in bengalese finch song development (Sasahara et al, 2015, not cited). Most recently, an electrophysiological study by Okubo et al, (2015, not cited) described this duplication and differentiation strategy at the mechanistic level of the 'splitting' of HVC chains.

1.2. Past studies showed a diversity of learning strategies, some of which actually superficially appear consistent with global alignment for syllable sequencing. Specifically, Liu et al showed that many birds appeared to exhibited a 'motif-level' learning strategy in which at the onset of plastic song, at the very early stage of learning, there are already a diversity of sequentially ordered syllables that gradually evolve to align with the entire tutor sequence. Okubo et al. also observed this phenomenon in rare cases (n=2 of 35 birds), and explained the strategy at the level of HVC chains. These 'motif-learners' (Extended Figs 8-9 of Okubo et al.) exhibited long HVC chains in which every syllable in the nascent motif had at least one neuron that was shared with another syllable at similar latencies, consistent with the view that all of the syllables arose from the splitting of a common ancestor. In this case, the duplication event had already occurred and the early sequence of unique protosyllables was eligible for motif-level global alignment. The main point here is that even in the case of 'motif-learners' there is substantial evidence that duplication is of fundamental importance in song learning, seemingly inconsistent with global alignment as a process that guides all stages of song learning.

1.3 Third, a previous study showed that when birds learn to produce target ACB from ABC new transitions are added in a stepwise fashion (Lipkind et al, 2013). If the birds were using global alignment then they would change the phonology of the current syllable number to match the new target. They would not need to introduce new transitions, either in one shot or step by step.

1.4 Finally, there are many species of songbirds that do not sing a definite sequence of syllables. Such non-deterministic singers (such as bengalese finches, nightingale, starlings, willow warbler) produce variable repetitions of different syllables (trills), making the global alignment hypothesis for song learning untenable.

Summary of point 1, action requested in revision:

In revision, can the authors please address how their theoretical framework and behavioral results fit with the duplication and differentiation strategy of song learning, which already seemed grossly incompatible with both the global alignment hypothesis and with the musical chairs metaphor?

2. Clarification on the specificity versus generality of the findings

2.1 The two-tutor training paradigm necessarily captures a late stage of song learning when syllable duplication appears no longer able to occur. Specifically, in previous studies (cited above) splitting and duplication occurred at the early plastic song stage, between dph 40-55. In the present study, birds are being trained to incorporate modified syllables and/or sequences at a later stage (dph >60). Syllable duplication and subsequent differentiation seems incompatible with the musical chairs metaphor, unless this metaphor is only meant to apply to the specific stage of learning where syllables have already been differentiated. This is relevant to Fig 5, for example, when there is incorporation of a call into a new version of syllable B. It is difficult to understand this result in the context of previous body of work showing that in younger birds a given syllable A can split and then become syllables X and Y (e.g. Tchernichovski et al, 2001; Okubo et al, 2015).

2.2. In the section "Differentiation in situ" from Tchernichovski et al, 2001; a very similar question of learning both phonology and sequence is addressed at the level of individual syllables. Here, they found that for matching of sub-syllabic acoustic features, the global alignment hypothesis is actually strongly supported. Specifically, within individual syllables there were constraints on sound

translocation not observed in the present manuscript for syllable re-sequencing. For example, in Fig 5C of Tchernichovski et al 2001, a target syllable with a harmonic stack transitioning into a broadband downsweep, is imitated from a source with a broadband segment. Rather than incorporating an untutored harmonic stack that the juvenile could sing at a different time in the song, the bird implements 'in situ differentiation' by evolving the broadband note into a stack. In the language of that paper, "this observation suggests that the laborious in situ differentiation of the harmonic sound was necessitated by constraints that hinder sound translocation and did not arise from a lack of 'appropriate raw material' for generating this sound." In this case, the raw material was a harmonic stack at the wrong 'time', after the syllable, yet the bird did not simply translocate it – instead it re-learned a new stack from broadband note of the first segment of the syllable. This past finding demonstrates that zebra finches may in fact use the global alignment strategy for sub-syllabic matching.

Altogether, these past studies show that birds appear to use different strategies at different song timescales and different stages of learning. This could actually be considered in the framework of computational complexity that the authors propose. Because a zebra finch syllable is itself a sequence of a large number of sub-syllable gestures, calculating the optimal translocations could be computationally prohibitive, and as a consequence finches simply learn sub-syllabic features by global alignment. In contrast, because they have limited number of discrete syllables they can learn new transitions efficiently.

Summary of point 2, action requested in revision:

In revision, can the authors please address how their theoretical framework would operate at the level of sequencing sub-syllabic gestures (e.g. Tchernichovski et al. 2001), including the problem of in situ differentiation and matching intra-syllable gestures (e.g. on ~5 ms timescale) to specific acoustic targets?

Minor Points

1. The authors use the words gestures, elements and syllables loosely and interchangeably, but the findings of this paper appear only to apply to discrete syllables.
2. In Fig. 2B, the transition from A to C+ seems to occur at 18 days after switch. But in Fig. 2C the transition in the same bird occurs at 22 days. These results are seemingly incompatible.
3. In Fig 2B there is a larger gap between A and C+ at about 18 days after the switch (early after the stability of the new transition). This looks to be real and potentially interesting. Are new transitions associated with longer gaps?
4. In Fig 2D, there appears to be something 'special' about 0.5, as evidenced by the apparent 'line' that would represent a steep falloff after 0.5. Can the authors please clarify?
5. It is unclear in Fig. 3 if the pitch distributions for syllable B are for all renditions of syllable B or if they are only considering those renditions of syllable B that occur after syllable A. Do any B's come after C and, if so, do they exhibit distinct pitch distributions?
6. The pitch oscillations (e.g. in Fig. 3D panel 2) are interesting but appear inconsistent with purely greedy phonology learning. Can the authors please clarify?

Reviewer #2 (Remarks to the Author):

This study addresses how the zebra finch, a vocal learning songbird species, copes with the need to learn both a vocabulary of vocal gestures and to organize these gestures into a temporal sequence, in order to match a target model. This important question has relevance for understanding how animal groups acquire their vocal repertoires by imitation, and for speech and language acquisition in humans. The study takes advantage of the detailed knowledge on song structure and learning in zebra finches, and the evidence indicates that these birds can separately correct spectral vs temporal mismatches. They appear to prioritize correcting spectral mismatches of individual syllables over the temporal sequence, when given tasks that in a way forces them to make a choice. The findings provide new insights, the methods are adequate and the evidence seems convincing. While this is a significant contribution, there are some issues with data interpretation, and the authors jump to some conclusions without enough data. These aspects need to be better addressed and some statements toned down.

This is a study of zebra finches, not of birds in general, as some passages seem to imply (e.g. "To test which strategy birds employ to learn their song..."). It is unclear whether the findings apply to all vocal learning birds, even less to birds in general. A very large number of bird species are thought to be vocal learners and their songs have very different acoustic structures. Little is known about how these song patterns emerge, thus one should be cautious in assuming that the rules being uncovered here have general applicability. Examining at least another species with a different song structure (e.g. motifs with repeated syllables, different motif structure or syllable sequences, etc) would have helped, but this point is not even discussed.

The authors suggest that the rules uncovered might apply to motor learning in general, and provide an analogy with learning to play tennis, but it is unclear whether this is a valid comparison. What would be the equivalent in terms of discrete gestures in this non-vocal motor skill? What is the evidence that such gestures are separate elements that could be learned and/or produced separately? This comparison seems to be quite a stretch. In contrast, there is no discussion or even mention of what might be the general learning constraints that non-avian vocal learning animals might face in acquiring their vocal repertoires. Because arguably similar auditory-motor constraints are involved, such cases might be of more relevance here than discussing the acquisition of non-vocal motor skills.

The experimental paradigm of sequential learning of songs that differ by a discrete element is clever, as it allows for dissecting the capabilities and constraints of the vocal learning apparatus of finches. However, it does not exactly reflect what a young zebra finch naturally encounters or the tasks it has to solve. At the start of vocal learning juveniles have not yet acquired their own song, and normally they may not need to modify an acquired song to match a moving target song. So while the study may show what finches are capable of doing, and thus illuminate some features of their learning apparatus, it does not necessarily show the normal pathway that vocal learning follows.

In some experiments, the generality of some conclusions is unclear. In expt. 3, it is interesting that different birds shifted the modified target syllable in different directions. Lowering the pitch of syllable B would have met the demands of both the spectral and temporal matches, however some birds chose an upward shift for this position. Could this be due to a salience effect of the last (in this case 4th) target syllable? Would the same be seen if the syllable that challenged the temporal sequence were at a temporally more neutral position of the motif? This is not to suggest that the authors need to test all possible interchanges, which might require using a longer motif with more syllables, but just to indicate that even with all manipulations provided, only a subset of possible combinations were tested,

thus the conclusions should be tempered as well.

Still regarding this experiment, the example on Fig. 3d, 2nd graph, is intriguing. This bird seems to be able to almost instantaneously switch the pitch of syllable B, suggesting that it has both versions of this syllable in its repertoire, but is trying to find which one would result in the best temporal sequence match. While somewhat anecdotal, this seems to illustrate well the point that syllables may be treated as independent elements that can be modified separately, and/or used in multiple combinatorial arrangements. In this experiment, however, it is unclear whether the birds eventually learned or not a motif with 4 syllables, as in the second target song. If not, why not? If a 4th syllable was eventually added, what signal did it have, the correct one? Was this a sequential process?

The issue of possible high salience of the last syllable is partially addressed in expt 4, since in that case the 4th syllable seems to shift towards an internal (2nd) syllable. Task 4.2 seems less convincing in this regard, since one cannot know for sure if the shift of either B syllable was towards the 2nd or 4th target B syllables, in both cases the direction of the shift would be the same and what matters more is the size of the shift. What is unclear is whether the birds would continue shifting their syllables beyond the period shown here. Did the birds crystallize their songs, or did they show further pitch shifts beyond the period shown?

Specific points:

Abstract:

The authors comment: "and searching for the optimal transformation quickly becomes computationally intractable." From the bird's perspective, this seems irrelevant, it solves the problem whether or not a clear computational solution is immediately clear. This issue reflects more a limitation in current methods than a biological problem per se. This statement should probably not be in the Abstract, or at least should be modified.

In: "to correct conflicting phonological and sequential mismatches in song syllables" it will be unclear to the reader that the mismatches are between the already acquired song and a second target model; perhaps the paradigm could be better explained here.

The authors speculate: "and could perhaps be a generic solution in the evolution of motor learning mechanisms." This is quite a stretch, it would be more relevant to speculate that this may be a generic solution to vocal learning mechanisms.

In several places the authors refer to 'birds' instead of zebra finches, or even songbirds, they should be more accurate throughout the paper. If referring to the birds in the present study, they should use 'the birds' instead of 'birds'.

The authors state: "Once birds copied it...", but it is unclear what was the criteria for learning at the time when the target song was switched. Did the birds need to reach a minimum level of imitation of the first target song? How accurate or variable was the birds' imitation of that first song? Since there is considerable learning variation at any given age, were all birds at a somewhat uniform stage in the learning? This is important to better understand until what age or stage the birds would be capable of the vocal plasticity being shown here.

In: "one transition at a time (Fig. 2c)...", this passage is unclear. The graph indicates that the birds started singing some syllable transitions (BA, CB) not found in the initially learned song well before they started singing the AC transition, which is necessary for completing the new acquired sequence. This probably means that the birds started singing more varied types of motifs, or incomplete

renditions with 2 syllables only, but this is not clear from the text. This point should be more explicitly stated and explained.

In the literature the term 'acoustic' is usually meant to refer to both spectral and temporal features. Here the authors seem to be using it as equivalent to the spectral aspect only. This should be clarified, or perhaps switched to spectral, structural, or frequency-related.

In: "even when phonological errors are almost as small as production variability", the authors introduce a concept that had not yet been discussed in the paper, namely that there is measurable variability in vocal production patterns. For this to make sense, some definition of the naturally occurring production variability and how it is measured should be presented, and then compared in terms of amplitude to the measured 'phonological errors'.

Minor:

We tested AN additional nine birds

A B C B+1 ↔ A B+2 C B-1

This notation in the pdf gives the impression that there is a break in the sequence, the letters should be moved closer together for clarity.

Reviewer #3 (Remarks to the Author):

In this article, Lipkind et al. address the broad question of how animals modify their learned motor sequences to match both sequence information and structural information. They address this question by using the zebra finches as their model system. Specifically, they train juvenile zebra finches to produce a particular sequence of syllables (source) and then once they have learnt this, they switch to a new target that typically involves both a sequence change and a phonology change (pitch shift). Using a few different types of target sequences, they show that zebra finches change the phonology first and this phonology change results in a sequence error that is eventually corrected in a subset of birds. Using this data, they suggest that birds prioritise phonology learning over sequence learning, thereby solving what would have otherwise been a computationally intractable task – learning both phonology and sequence simultaneously. They also suggest that these two might represent two separate modules for learning sequences.

Overall, the paper addresses an important and interesting question. The authors address this question very nicely with appropriate experiments and analysis. Their claims are well supported by the data. Their results are novel and are likely to be of interest to the songbird community and to a wider audience. For the most part, the paper is well written and easy to follow. However, I have a few minor concerns that are listed below.

1. I am assuming that under natural conditions, zebra finches do not decide to switch to a different song midway through their learning. Therefore, this task could be a little un-natural for them. Clearly young birds cope with this and do learn the new target song and this is definitely a nice way to address the original question of how sequence and structure are both learned. However, whether similar processes are involved in natural learning is not fully discussed by the authors. The authors briefly discuss other literature related to song learning in juveniles in the Results section, but they do not fully discuss possible reasons for the difference in strategy. After all, one prominent strategy for song learning (Liu et al, 2004 - ref. #22) shows that young birds can use a "motif" strategy where

they set down a temporal sequence and slowly modify the acoustic structure. The authors could expand on their discussion of reasons for the difference in strategy that they observe.

2. They cite an older study (Lipkind et al. 2013) as evidence that syntax learning is not age-dependent, but a large number of birds in that study also did not fully learn syntax changes. It appears like syntax learning capabilities could be limited by age. Did the authors observe any correlation between the day on which playback was switched to the target motif and the extent of syntax learning?

3. In this study, the authors use pitch learning as a proxy for learning acoustic structure. Zebra finch syllables have varying degrees of acoustic complexity and I wonder if modifying pitch can be generalized to modifying acoustic structure in general? This is again a potential difference between normal song learning (earlier studies), where birds have to learn more complex acoustic structure (which can sometimes involve pitch changes too, but not the only change). Pitch changes are obviously easier to track and follow from an experimenter's perspective, but do the authors think this would generalize to all kinds of acoustic changes?

4. In the Methods, the authors mention that they switched to playbacks of the target song once the birds learned the source song. Can the authors specify the criteria that they used to assess whether birds had learned the source song or not?

5. Finally, in the abstract, the authors state " ... resulting in unnecessary sequence errors that were later corrected". Given that a large number of birds do not fully correct sequence errors and do not achieve perfect imitation of the target sequence, this seems inaccurate to me. It could be changed to " ... unnecessary sequence errors, some of which were later corrected" or something to that effect emphasizing that all errors were not corrected.

We thank the reviewers for their helpful comments, all of which we have addressed. We believe the revised MS is significantly improved. In addition to revisions addressing the reviewers' comments, we made a small revision to the Discussion (main text, pages 7-8, lines 292-294) and Supplementary Text (Supplementary Information, page 5, lines 151-160) to point out a similarity between zebra finches' strategy of matching song vocabulary and a state of the art method for evaluating document similarity (word mover's distance), of which we were not aware before. All revisions in the main text and Supplementary Information are highlighted.

Response to reviewers' comments (reviewers' comments are in italics; authors' responses are in plain text):

Reviewer #1 (Remarks to the Author):

I previously reviewed this paper for a different journal. While very little has changed in this re-submission, I outline below why I think this is an extremely important contribution to the birdsong field and to sensorimotor learning more generally, and is appropriate for publication in Nature Communications.

We thank the reviewer for these remarks.

There are several issues that I raised previously that I think could be addressed in discussion without further experiments. As I specify below, the main issue is tailoring the conclusions to the specific experiments and better contextualizing the findings with past work, specifically on sub-syllable matching and on duplication/differentiation strategies for learning.

We have addressed all of the issues raised by the reviewer, as detailed below.

Lipkind et al identify algorithms that zebra finches implement as they learn to match motor sequences to sensory targets. This is an important question that has the potential to generalize to broad classes of sensorimotor learning. One outstanding question is how animals deal with the problem of simultaneously matching both the structure and sequencing of targets. To address this issue, Lipkind et al build on previous work where they expose birds to an initial tutor song and observe how birds modify their song after exposure to a second tutor. In this study, they first develop a theoretical framework that formalizes the computational challenges of phonological and sequence learning. To identify strategies that the bird employs to reduce these challenges, they specifically devise tasks that cause direct 'conflict' between syllable phonology and syllable sequence. In this two-tutor paradigm, many birds prioritize phonology matching at the expense of sequence matching, in violation of predictions of a global alignment hypothesis of error correction. The main strength of the paper is that the training sets are very creatively designed to specifically pit phonology versus sequence. For example, the design of imitation task 4 was particularly clever and showed very nicely that the birds can indeed be deaf to sequence (and acoustically greedy); they did not choose to move a pitch of the syllables a little more (presumably a small increase in cost), with the benefit of keeping the sequence right. The core observations are clearly presented and for the most part well analyzed and show that under certain conditions many zebra finches exhibit a clear tendency to prioritize phonology matching, even at the cost of sequence matching. A second strength is that the supplement provides a much-needed formalism for the problem of target-matching that will be useful for the birdsong field.

While the big concept - that global alignment is not always deployed - is generally consistent with past work, the present paper provides a more definitive and rigorous demonstration of the idea. One main issue is I would like to see addressed, at least in discussion, is how their findings can generalize to the sequencing of sub-syllable elements into syllable targets, to earlier developmental stages, and to different species that as adults have variable sequencing. I outline these concerns below.

We have updated the Introduction, Results, Discussion and Supplementary Text to address the reviewer's concerns. See detailed responses below.

1

1.1. The main experimental finding was that the birds were able to move the pitch out of context, in direct conflict with a global alignment model of error correction. While the present paper demonstrates this more

elegantly than past work, it was already suggested by previous work that birds can do out-of-context comparisons and that the global alignment hypothesis was not universal.

We revised the introduction to include a description of previous studies suggesting that songbirds may not be using a global alignment algorithm for song learning, and added the omitted references pointed out by the reviewer (main text, page 2; lines 70-78; see revised segment on page 4 below). We would like to note, however, that experimentally refuting the hypothesis of global alignment is merely the starting point of our study (tasks 1 and 2, results). We then go on to demonstrate the complete independence of phonological error computation of any sequential context beyond the single syllable (tasks 3 and 4), pointing to the existence of two independent modules for learning syllable vocabulary and sequence; and finally we elucidate a "musical chairs"-like algorithm for fully matching a given syllable vocabulary in the absence of sequence information (tasks 3 and 5). To the best of our knowledge, these findings and conclusions were not demonstrated, or even suggested, by previous studies.

For example Immelman (1969, not cited) found that syllables imitated from a model song appear early in development before the appropriate serial order of syllables was apparent, showing both that phonology is prioritized and that it can be learned independent of sequence.

The early qualitative observation by Immelman indeed suggests that phonology can be matched outside the global context of the target song, and we now cite it (main text, page 2, lines 72-73).

Liu et al (2004, cited) showed that zebra finches (including brothers) use a variety of strategies to learn the same tutor song. Early in development (<55 dph) many birds serially repeat a single syllable; this single syllable subsequently duplicates and independently differentiates into multiple 'descendant' syllables. This same duplication and differentiation strategy has been demonstrated in several papers: Tchernichovski et al (2001, e.g. Fig 5), in the 'protosyllables' described by Aronov et al. (2011, not cited), and in bengalese finch song development (Sasahara et al, 2015, not cited). Most recently, an electrophysiological study by Okubo et al, (2015, not cited) described this duplication and differentiation strategy at the mechanistic level of the 'splitting' of HVC chains.

We omitted to mention the phenomenon of duplication and differentiation of early syllable precursors in our MS. The reviewer's comments made us realize that this was an oversight, and we have revised the MS accordingly (main text, page 2, lines 73-75; page 4, lines 154-159; page 5, lines 198-200; pages 9-10, lines 337-374; see also our responses to the reviewer's comments on early development below, points 1.2 and 2.1, and Summary of point 1). Prototype duplication and differentiation is consistent with the possibility of non-global alignment strategies, and we now include this developmental strategy in the revised introduction and have added the missing references (main text, page 2, lines 73-75). However, the evidence provided by the studies cited by the reviewer is somewhat mixed, regarding the question of how duplicating and differentiating proto-syllables *select their targets*. In fact, Liu et al (2004) interpreted their findings as indicating global alignment learning strategies (see more on that in response to point 1.2 below). We are now addressing this point in the discussion (main text pages 9-10, lines 337-374; see also response to point 2.1 below). Another issue is that it is not clear if and how the duplication/splitting of syllable precursors is related to target assignments; it could be that splitting is triggered by target assignments (i.e., that a proto-syllable is "pulled apart" by two different targets into two daughter-syllables); or it could be that splitting occurs by some other mechanism, before targets are assigned, providing precursor syllables that can then be assigned to (and differentiate towards) diverse target syllables. We elaborate more on this point in response to R1's point 2.1 below.

1.2. Past studies showed a diversity of learning strategies, some of which actually superficially appear consistent with global alignment for syllable sequencing. Specifically, Liu et al showed that many birds appeared to exhibit a 'motif-level' learning strategy in which at the onset of plastic song, at the very early stage of learning, there are already a diversity of sequentially ordered syllables that gradually evolve to align with the entire tutor sequence. Okubo et al. also observed this phenomenon in rare cases (n=2 of 35 birds), and explained the strategy at the level of HVC chains. These 'motif-learners' (Extended Figs 8-9 of Okubo et al.) exhibited long HVC chains in which every syllable in the nascent motif had at least one neuron that was shared with another syllable at similar latencies, consistent with the view that all of the syllables arose from the splitting of a common ancestor. In this case, the duplication event had already occurred and the early sequence of unique protosyllables was eligible for motif-level global alignment. The main point here is that even in the case of 'motif-learners' there is substantial evidence that duplication is of fundamental importance in

song learning, seemingly inconsistent with global alignment as a process that guides all stages of song learning.

We agree with the reviewer's view that cases of "motif-learning" may result from an earlier unobserved stage of proto-syllable duplication, and we now discuss the evidence on and interpretation of the motif learning strategy in the MS (main text, page 9, lines 355-363; see response to point 2.1; and also Reviewer 3's point 1 on page 18 below). While it is possible that at some stages of song learning zebra finches are capable of employing global alignment, we do not think that "motif-learning" trajectories necessarily imply a global alignment strategy, especially if they are rare. The reason is that an acoustically greedy (or "motif-deaf") learning strategy, does not preclude rare cases of "motif learning", because sometimes greedy vocabulary matching can lead to correct sequence matching by chance, (e.g. when a bird sings something like ABCB⁻¹ and his target is AB⁺¹CB⁻²). Therefore, in our view, the question of whether global alignment is employed at early stages of song development boils down to how frequently motif learning is observed. Okubo et al. found "motif learning" to be rare, which argues against global alignment early on; Liu et al., found motif learning to be common, and also reported that the prototype duplication cases they found were in agreement with global alignment (in their words: "In the "serial repetition" strategy ... different syllables from the final imitation emerge, through modification, from repetitions of the single syllable; they do so already in the order, relative to each other, in which they will appear in the adult song"; see also their Figure 2). Therefore, although we agree with the reviewer in thinking that global alignment is probably not employed in either early or late development, given the results of Liu et al, the possibility of global alignment early on cannot be ruled out at present. We believe that the question of target matching strategies very early on may need to be resolved on the neural level, perhaps by recording from HVC premotor neurons both during singing and during playbacks of the target song. We now make this point in the discussion (main text, pages 9-10, lines 363-374; see more about this in our response to point 2.1 below).

1.3 Third, a previous study showed that when birds learn to produce target ACB from ABC new transitions are added in a stepwise fashion (Lipkind et al, 2013). If the birds were using global alignment then they would change the phonology of the current syllable number to match the new target. They would not need to introduce new transitions, either in one shot or step by step.

We now mention this point in the revised part of the introduction (main text, page 2, lines 76-78). But still, our Lipkind et al 2013 paper did not show conclusively that the global alignment hypothesis is incorrect. All the vocal changes performed by birds in that study (adding individual target bigrams to their repertoire) increased the match with the target both locally and globally. Birds were not presented with the option to make vocal changes that improve the local but not the global match with the target, as in the present study. Note also that the task in Lipkind et al 2013 (ABC → ACB), unlike the tasks in the present study, was a "pure" syntax learning task. Namely, it *could* be accomplished without any changes in phonology. Thus, this task was not designed to test for global alignment as a solution to the full song-learning problem (sequence-to-sequence matching). In addition, the results of Lipkind et al 2013 are perfectly consistent with the hypothesis of piecewise alignment across syllable chunks (e.g., across bigrams), which is the n-gram hypothesis in Figure 1c that our current study refutes.

1.4 Finally, there are many species of songbirds that do not sing a definite sequence of syllables. Such non-deterministic singers (such as bengalese finches, nightingale, starlings, willow warbler) produce variable repetitions of different syllables (trills), making the global alignment hypothesis for song learning untenable.

We revised the MS to point out that many songbird species sing variable sequences, arguing against the global alignment hypothesis (main text, page 2, lines 75-76). We now also mention this fact in the discussion, regarding the role of non-global alignment algorithms in learning large repertoires of vocal sequences (page 10, lines 384-386; see our response to R2's comment on pages 11-12 of the rebuttal).

To summarize our response to the reviewer's comments so far:

We have updated the introduction to include previous findings that suggest (or are consistent with) the possibility that birds do not use global alignment as a strategy for matching their song to their target (page 2; lines 70-78 ;). These include: early evidence for phonology appearing before syntax during development; the phenomenon of duplication and differentiation of proto-syllables; the fact that birds can accomplish a pure sequence rearrangement task (ABC → ACB) without changing phonology; and the existence of variable song

syntax in many songbird species. We are now citing the omitted studies pointed out by the reviewer. The revised paragraph now reads (newly added part is highlighted; reference #s refer to main text):

A possible way to reduce computational complexity could be to optimize one aspect of the task, while ignoring the costs of the other. At one extreme, the task could be reduced to assigning each syllable in the bird's song to the temporally corresponding syllable in the target song (Fig. 1c left). Such strategy would minimize sequence rearrangements, at the cost of possibly large phonological adjustments. **Although this hypothesis has not been directly tested, a number of previous findings suggest that songbirds may not be using global alignment between song and target as a learning strategy. These include the observation that individual syllables are recognizable in developing zebra finch song before the correct sequence is apparent²⁸; the existence of an early developmental phase in which repetitions of a single "proto-syllable" differentiate towards multiple targets^{22,24,25,29,30}; the fact that many songbird species perform variable syllable sequences as adults (e.g. nightingales, starlings and Bengalese finches); and the ability of zebra finches to match a target exclusively through syllable rearrangements, without changing phonology²⁶. An alternative strategy, therefore, could be to assign song syllables to target syllables in a manner that minimizes phonological distances, while ignoring combinatorial distances (Fig. 1c middle). Such phonological greediness would increase the number of ensuing sequence changes and thus the overall sequencing cost²⁶. An intermediate strategy could be to seek a trade-off between minimizing structural and temporal errors, for example by independently matching parts of the song sequence (such as phonology in bigrams or trigrams²⁷) to parts of the target sequence (Fig. 1c, right).**

We copy here the reviewer's point 2.1, as it is relevant to the action requested for point 1:

2.1 The two-tutor training paradigm necessarily captures a late stage of song learning when syllable duplication appears no longer able to occur. Specifically, in previous studies (cited above) splitting and duplication occurred at the early plastic song stage, between dph 40-55. In the present study, birds are being trained to incorporate modified syllables and/or sequences at a later stage (dph >60). Syllable duplication and subsequent differentiation seems incompatible with the musical chairs metaphor, unless this metaphor is only meant to apply to the specific stage of learning where syllables have already been differentiated. This is relevant to Fig 5, for example, when there is incorporation of a call into a new version of syllable B. It is difficult to understand this result in the context of previous body of work showing that in younger birds a given syllable A can split and then become syllables X and Y (e.g. Tchernichovski et al, 2001; Okubo et al, 2015).

It is true. We did not observe duplication and differentiation as a target matching strategy. In our experiments, when a syllable was offered two acoustically close targets (syllable B in tasks 3 and 5), it shifted to match only one of the targets, and the other target was matched by a separate precursor (usually a call). We did not observe cases where syllable B "split" to match both targets B⁻ and B⁺ (except for a possible unsuccessful attempt in one bird, see our response to the reviewer's minor point 5 below). We now point this fact in the relevant sections of the results (main text, page 4, lines 154-158; and page 5, lines 198-200). Therefore we agree with the reviewer that we cannot reliably extend our model to account for syllable duplication and differentiation, and cannot at present rule out the possibility that at early developmental stages, when prototype duplication occurs, birds may employ a different target matching strategy to the one we discovered. We now point this out in a new section added to the discussion ("Target assignments in early vocal development", main text, pages 9-10, lines 337-374; see also the new section on pages 5-6 below).

However, duplication and differentiation may still be compatible with our musical chairs model, including its three components: 1) acoustic greediness (error is computed with respect to acoustically similar targets regardless of their temporal position; constraints C1 and C2, Supplementary Text, page 4); 2) winner-take-all target selection (error is not averaged across equidistant targets, but only one target is selected; constraint C2); and 3) competitive target matching (error is not computed with respect to "occupied" targets; constraint C4). However, duplication and differentiation is also consistent with alternative assignment algorithms. Specifically:

1. Duplication and differentiation is consistent both with acoustically greedy and with context-dependent target assignment algorithms. Acoustically greedy assignment would imply that syllable prototypes differentiate towards acoustically close targets (with specific assignments perhaps determined by random acoustic variation among prototype renditions). This would mean that two adjacent renditions of a proto-syllable could differentiate towards two *non-adjacent syllables in the target song*. On the other hand, a context dependent assignment algorithm would mean that syllable prototypes must differentiate towards target syllables at corresponding temporal positions.

2. Duplication and differentiation is consistent with winner-take-all target selection (that would imply that a prototype presented with two equidistant targets would split to match both); but also with averaging errors across targets (that would imply that a prototype presented with two equidistant targets would not shift towards either, since the computed error would be zero). Interestingly, if an acoustically greedy strategy is employed to assign prototype renditions to targets (point 1 above), a winner-take-all mechanism could be useful in stabilizing assignments, and avoiding fluctuating assignments due to high variability of early song.

3. Duplication and differentiation is consistent with competitive target matching (in such a case, a prototype presented with several targets, some of which are occupied and some vacant, will split to match only the vacant targets); and also with redundant target matching (in that case, a prototype will split to match targets regardless of whether they are vacant or occupied).

Another crucial question to consider here is: are targets assigned before or after splitting? The former scenario can be imagined as two (or more) targets “pulling apart” a single proto-syllable into several daughter-syllables, and this process may be restricted to an early developmental stage (~<55 dph). The latter scenario can be seen as a process that generates “raw material” for subsequent target assignments. Thus, the reason we did not observe splitting of syllable B to match both B- and B+ could be because at the age at which our experiments were conducted, new targets can no longer trigger a syllable to split (scenario 1), or it could be that syllable B could not split because it was already assigned to a target (B), and splitting occurs only in “unassigned” proto-syllables (scenario 2).

This question is important because in the case of scenario 2, splitting of proto-syllables is completely irrelevant to the question of syllable-target assignment algorithms; but in the case of scenario 1, one needs to consider the possibility that target assignment algorithms employed in early proto-syllables differ from those employed in later development.

The challenge in experimentally testing these possibilities lies in being able to identify assignments between proto-syllables and their target syllables as early as possible, even before duplication and differentiation are detectable behaviorally. Okubo et al, made a big step in this direction by tracking the activity of premotor neurons in HVC during singing at the subsong and proto-syllable stages. It has recently been shown that in young zebra finches, HVC premotor neurons exhibit precise firing not only during singing, but also when listening to the tutor song, potentially providing an instructive signal for song development¹. Therefore, a possible future solution could be to combine tracking of neuronal activity of HVC projection neurons during singing (as in Okubo et al. 2015) and during tutor song playbacks (as in Vallentin et al. 2016), which could uncover how syllable-target assignments are formed in early development.

To summarize:

We added a section to the discussion (main text, pages 9-10, lines 337-374), where we explain the limitations of our experimental method for inferring assignment strategies during early development, discuss the predictions of our model for duplicating and differentiating proto-syllables, and suggest a possible way to test them. The new part also includes a short discussion on the evidence for “motif learning” in early development (see our response to the reviewer’s point 1.2 above, and also Reviewer #3’s point 1 on this on page 18 below). Note also changes in terminology according to Reviewer #2’s suggestion below (the term “acoustic” was replaced with “spectral” or “phonological”).

The new section reads:

Target assignments in early vocal development

Our serial tutoring paradigm is confined to mid and late stages of song development, at which the early phenomenon of duplication and differentiation of syllable prototypes to match multiple targets^{22,24,25,29,30} is no longer observed. In our experiments, when a syllable was offered two spectrally similar targets (tasks 3 and 5), only one of those targets was selected - the syllable never “split” to match both. We therefore do not have direct evidence as to what target assignment algorithm may govern the duplication and differentiation of early proto-syllables, and even on whether proto-syllable splitting is related to target assignments at all or occurs prior to targets being assigned. We cannot rule out the possibility that the “musical chairs” constraints we discovered might not (or not fully) apply to early song development. If our model applies to earlier stages of song

development, this would mean that duplicated proto-syllables differentiate towards targets chosen by a phonologically greedy and competitive algorithm. In such a case, adjacent renditions of a proto-syllable, would not necessarily differentiate towards temporally adjacent syllables in the target song, but would instead differentiate towards targets according to spectral similarity. Since the spectral structure of early proto-syllables varies considerably across renditions^{23,29,37}, an interesting possibility is that target assignments may be initially determined by random spectral variation among proto-syllable renditions, and later reinforced by competitive musical-chairs-like constraints.

A previous study reported variable learning strategies in young zebra finches, including a “motif-learning” strategy²², where syllables in mature zebra finch song develop from very early precursors already arranged in the correct sequential order. This suggests that zebra finches are capable of employing a global alignment target matching strategy (Fig 1c, left) at early stages of their development. Another study of early development observed rare cases of motif learning²⁵, and showed that they originated from an earlier stage of duplication and differentiation of a single proto-syllable precursor, which is consistent with a non-global phonologically greedy strategy (since phonologically greedy target matching could lead to correct sequence matching by chance in rare cases). The challenge in resolving the question of target assignment strategies during early development, and in particular of testing the predictions of our model for early syllable precursors, lies in being able to identify specific assignments between variable proto-syllable renditions and their targets as early as possible. A recent step towards achieving this goal²⁵ was recording the activity of neurons in the premotor song nucleus HVC during singing in young zebra finches, allowing tracking of the process by which a single proto-syllable splits into two or more neurally distinct syllable precursors. Interestingly, HVC premotor neurons have been shown to exhibit precise auditory responses to the target song in young songbirds^{32,62}, possibly providing an instructive signal for song development³². Therefore, combining recordings from HVC during singing in juveniles with recordings of auditory responses to the target song in the same neurons may, in the future, allow to track the process by which targets are assigned to early syllable prototypes.

Summary of point 1, action requested in revision:

In revision, can the authors please address how their theoretical framework and behavioral results fit with the duplication and differentiation strategy of song learning, which already seemed grossly incompatible with both the global alignment hypothesis and with the musical chairs metaphor?

We agree with the reviewer that our study cannot be reliably extended to early developmental stages at which prototype duplication and differentiation occurs, and we revised the MS to clarify this point. We are not sure we fully understand the reviewer’s claim that duplication and differentiation “seems grossly incompatible with both the global alignment hypothesis and with the musical chairs metaphor”. Certainly, whether our musical chairs model applies to early developmental stages of duplication and differentiation is an open question, but we now offer some predictions as to how duplicating and differentiating prototypes would behave under the constraints of our model, and suggest a possible way to test these predictions in the future (by parallel recording of premotor activity and auditory responses in HVC pre-motor neurons during early development).

To summarize, we have made the following revisions to address the reviewer’s comments in points 1 and 2.1:

1. We updated the introduction to present previous evidence suggesting that a global alignment strategy in song learning is not likely (page 2, lines 70-78).
2. We updated the results’ text dealing with tasks 3 and 5 to point out that we did not observe splitting of a syllable in the source song to match two syllables in the target song (page 4, lines 154-158; and page 5, lines 198-200).
3. We added a section to the discussion that explains the limitations of our method for studying early stages of syllable duplication and differentiation, the possible implications of our musical chairs model for target assignments during early song development, and a possible way to test them in the future (pages 9-10, lines 337-374).

2. Clarification on the specificity versus generality of the findings

2.1 *The two-tutor training paradigm necessarily captures a late stage of song learning when syllable duplication appears no longer able to occur. Specifically, in previous studies (cited above) splitting and duplication occurred at the early plastic song stage, between dph 40-55. In the present study, birds are being trained to incorporate modified syllables and/or sequences at a later stage (dph >60). Syllable duplication and subsequent differentiation seems incompatible with the musical chairs metaphor, unless this metaphor is only meant to apply to the specific stage of learning where syllables have already been differentiated. This is relevant to Fig 5, for example, when there is incorporation of a call into a new version of syllable B. It is difficult to understand this result in the context of previous body of work showing that in younger birds a given syllable A can split and then become syllables X and Y (e.g. Tchernichovski et al, 2001; Okubo et al, 2015).*

See our response to this point copied above (pages 4-6 of the rebuttal).

2.2. *In the section “Differentiation in situ” from Tchernichovski et al, 2001; a very similar question of learning both phonology and sequence is addressed at the level of individual syllables. Here, they found that for matching of sub-syllabic acoustic features, the global alignment hypothesis is actually strongly supported. Specifically, within individual syllables there were constraints on sound translocation not observed in the present manuscript for syllable re-sequencing. For example, in Fig 5C of Tchernichovski et al 2001, a target syllable with a harmonic stack transitioning into a broadband downsweep, is imitated from a source with a broadband segment. Rather than incorporating an untutored harmonic stack that the juvenile could sing at a different time in the song, the bird implements ‘in situ differentiation’ by evolving the broadband note into a stack. In the language of that paper, “this observation suggests that the laborious in situ differentiation of the harmonic sound was necessitated by constraints that hinder sound translocation and did not arise from a lack of ‘appropriate raw material’ for generating this sound.” In this case, the raw material was a harmonic stack at the wrong ‘time’, after the syllable, yet the bird did not simply translocate it – instead it re-learned a new stack from broadband note of the first segment of the syllable. This past finding demonstrates that zebra finches may in fact use the global alignment strategy for sub-syllabic matching.*

Altogether, these past studies show that birds appear to use different strategies at different song timescales and different stages of learning. This could actually be considered in the framework of computational complexity that the authors propose. Because a zebra finch syllable is itself a sequence of a large number of sub-syllable gestures, calculating the optimal translocations could be computationally prohibitive, and as a consequence finches simply learn sub-syllabic features by global alignment. In contrast, because they have limited number of discrete syllables they can learn new transitions efficiently.

We thank the reviewer for making this point, which we now address in the Discussion (main text, pages 8-9, lines 317-336) and Supplementary Text (supplementary information, page 6, lines 184-204). We agree that context-independent matching of song parts to acoustically similar parts of the target is not likely to be the sole strategy for learning an entire song – it has to break down at a certain point along the song time scale (e.g., it is extremely unlikely that birds independently match tiny ~5 ms-long song segments to the most acoustically similar pieces in the target, and then rearrange their positions to match the target sequence). Beyond this putative “breaking point”, we would expect that birds could no longer rearrange the positions of song parts and therefore must use a global matching strategy within these “unbreakable” song units. As the reviewer points out, given the findings of Tchernichovski et al 2001, a likely candidate for the transition point from context-free to context-dependent alignment is the song syllable. We have recently started testing this hypothesis with a dedicated serial tutoring task, which is analogous to task 1 in the current study (ABC → AC⁺B), but with B and C/C⁺ being two notes within a single syllable. Our preliminary results so far show that indeed birds do not rearrange the positions of sub syllabic elements, and mostly (except in one case) do not make greedy phonological adjustments of sub-syllabic notes out of context, but instead generate a new syllable from scratch (which is consistent with using global alignment within syllables). We feel that these findings deserve a separate manuscript, in which additional serial tutoring tasks will be used to elucidate the details of the algorithm used for matching sub-syllabic structure. At present, we have added a section to the discussion (“Target matching across song timescales”; main text, pages 8-9, lines 317-336) and the Supplementary Text (supplementary information, page 6, lines 184-204), which explain the theoretical grounds for expecting a switch from locally greedy to global alignment at some point within the song hierarchy, including the reviewer’s suggestion that such a switch could reduce computational complexity. We also point out the implications of

Tchernichovski et al., 2001 for expecting the transition from local to global alignment to occur at the song syllable boundary.

The new section in the discussion reads:

Target matching across song timescales

We found a phonologically greedy and context independent target matching strategy at the time scale of syllable vocabulary, but what happens at smaller time scales within individual syllables? Does context independence continue all the way to the level of the smallest controllable song segments (5-10 ms)⁶¹, or does it break down at a certain point along the song hierarchy? The answer depends on which are the smallest units within the song that a bird is able to rearrange. Namely, units that cannot be “moved around” with respect to each other must be learned in the appropriate context, since in such a case, sequence errors resulting from greedy learning could not be corrected. Given the laborious and time-consuming nature of sequence rearrangements²⁶, learning individual vocal gestures in a context-independent manner would be a highly inefficient strategy, as it would require a large number of positional rearrangements. In addition, resolving competitive conflicts between spectrally similar targets at such small timescales might be difficult, because many more conflicts would need to be considered, thus substantially increasing the number of computations to be performed. A more reasonable strategy would be to switch from context “deaf” to context dependent learning at small timescales. Previous evidence of in-situ differentiation of sub-syllabic elements during zebra finch development²⁴ indicates that the transition point from phonologically greedy to context-dependent target matching may be the song syllable. This would imply that zebra finches could not rearrange the positions of sub-syllabic elements within their song, a hypothesis that could be tested in the future with appropriate serial tutoring tasks.

The addition to the Supplementary Text reads:

Sub-syllabic notes

What is the smallest song unit to which our formalism applies? We deliberately called \mathcal{S}_j a song element and T_i a target element, implying these elements do not necessarily have to represent entire song syllables but could also represent sub-syllabic notes. In the following we discuss this possibility.

In our treatment of the song learning problem, we implicitly assumed that birds compute phonological error of a syllable by integrating over the errors in its constituent notes. Essentially, we assumed that birds compute the error of a syllable by globally aligning its notes with that of a template syllable. However, we have no evidence for this mini-version of global alignment. Thus, it remains to be explored whether birds can assign one of its syllable notes either to a note in a different syllable of the template or to a note in a different position within the same template syllable.

Although it will not be possible to resolve this issue without further experimenting, we imagine that our discovered assignment strategy cannot apply to ever smaller song units. Namely, at some point, there must be an overload to short-term memory arising from all these pairwise comparisons between song and template elements. It is therefore likely that assignment capabilities of zebra finches are limited to the syllable level and do not generalize to smaller song units below that level.

Summary of point 2, action requested in revision:

In revision, can the authors please address how their theoretical framework would operate at the level of sequencing sub-syllabic gestures (e.g. Tchernichovski et al. 2001), including the problem of in situ differentiation and matching intra-syllable gestures (e.g. on ~5 ms timescale) to specific acoustic targets?

We revised the Discussion and Mathematical Supplement to address the reviewer’s concerns regarding sub-syllabic matching strategies, see response to point 2.2 above.

Minor Points

1. The authors use the words gestures, elements and syllables loosely and interchangeably, but the findings of this paper appear only to apply to discrete syllables.

We corrected the wording to “syllables” in parts that deal with song learning, and we now use the term “gestures” only in parts that address the general question of learning complex motor sequences, for which the term “syllables” is too specific: the abstract (line 20), 2nd paragraph of the introduction (line 49), 4th paragraph of the discussion (line 310) and Fig 1a and c (lines 640, 642, 648-649 and 651).

2. In Fig. 2B, the transition from A to C+ seems to occur at 18 days after switch. But in Fig. 2C the transition in the same bird occurs at 22 days. These results are seemingly incompatible.

We corrected the error. We thank the reviewer for catching it.

3. In Fig 2B there is a larger gap between A and C+ at about 18 days after the switch (early after the stability of the new transition). This looks to be real and potentially interesting. Are new transitions associated with longer gaps?

We did not see a consistent tendency of new transitions’ being initially associated with longer gaps. We (some of us) have also looked into this question when analyzing data of a previous study (Lipkind et al., 2013), and found some cases where new transitions were associated with longer gaps, but this phenomenon was not consistent. It is indeed interesting, however, and justifies a detailed analysis of gap durations during sequence rearrangement tasks. But this is beyond the scope of the present study.

1. In Fig 2D, there appears to be something ‘special’ about 0.5, as evidenced by the apparent ‘line’ that would represent a steep fall off after 0.5. Can the authors please clarify?

This is a result of the stepwise acquisition of new syntax (Lipkind et al, 2013). New transitions between syllables are learned as discrete “steps”, where a transition is not performed at all for a long time, and then it appears and abruptly increases in frequency (see Figure 1d and g in Lipkind et al 2013). Therefore, for most of the developmental period the proportion of a target bigram (versus the corresponding source bigram) is either 0 (before the new bigram appeared) or ~1 (shortly after the bigram appeared), but relatively little time is spent at intermediate values. Here, we calculated the syntax match with respect to a pitch shifted syllable as the mean of the proportions of converging and diverging target bigrams (see methods; main text, pages 11-12 x, lines 457-466), and therefore frequent values of syntax match in Fig 2D were either ~0 (neither converging nor diverging target bigram appeared), ~1 (both converging and diverging target bigrams were learned), and 0.5 (one but not the other of the bigrams was learned).

5. It is unclear in Fig. 3 if the pitch distributions for syllable B are for all renditions of syllable B or if they are only considering those renditions of syllable B that occur after syllable A.

The distributions include all renditions of syllable B and the target syllable it shifted towards (including all stages in between), regardless of the context in which the syllable was performed. Renditions of B and its target were identified via clustering based on mean acoustic features of syllable renditions^{2,3} (see methods; main text, page 11, lines 434-448). In all cases, B shifted towards its target smoothly (see pitch trajectories in Fig 3b-d and Supplementary Fig 2a), such that all renditions in the process were identified as belonging to the same cluster.

Do any B's come after C and, if so, do they exhibit distinct pitch distributions?

In three out of the four birds trained with task 3 syllable B (and the target it shifted to) were performed solely after A, with a call differentiating to become the target performed after C (Fig 5c-f; Supplementary Figs 3a and 4c); However, in one bird, syllable B shifted to B⁺, and no other precursor differentiated to the missing target B⁻ (Supplementary Fig 2a and Supplementary Fig 3c; bird 3 in the previous version, renamed bird 4 in the current version, see below). Interestingly, in this bird some renditions of B/B⁺ were performed after C, though most were performed after A. Following the reviewer's question, we compared the pitch trajectories of B/B⁺ renditions performed after A and after C, and found that they gradually diverged such that the renditions after C were significantly lower than those after A on the last experimental day (although the difference was small, 25Hz, which is about half a semitone). This could be a late and unsuccessful attempt to “split” B⁺ to match the vacant target B⁻, but of course we cannot judge with any certainty from one bird.

We updated Supplementary Fig 3a, which now shows separate pitch trajectories for Bird 4 of B/B⁺ renditions after A and after C, and an additional plot of the daily mean pitch values in each context. The revised panel is shown below:

The revised legend of Supplementary Fig 3 reads (new parts highlighted):

Matching a vacant target with a vocalization initially performed outside of the song motif.

Developmental trajectories of experimental birds trained with tasks containing an extra syllable type in the target versus the source: task 3 (ABC → AB⁻CB⁺) (a) and task 5 (AB → AB⁺AB⁻) (b). Stack plots and pitch trajectories as in Fig 5 c and d (depicting Bird 1 of task 3). All birds except one (Bird 4 in (a)) matched the vacant target with a syllable type initially external to the song motif, usually a call. In bird 4 (a, bottom), the target B⁻ was not matched. Syllable B shifted to B⁺ in the “wrong” context (namely, after syllable A), but was also performed sparsely after syllable C; pitch trajectories in this bird are shown separately for renditions after A and after C (middle plots); right-most plot shows daily pitch means ± s.e.m. for renditions after A (black circles) and C (grey diamonds), showing a gradual divergence in pitch (756±1.6 Hz after A versus 731±1.9 after C on last experimental day; p<0.00001, t-test). This could potentially result from incomplete “splitting” of B⁺ into two syllable types.

Note that due to changing the order of presentation in Supplementary Fig 3a we switched the labels of two birds with respect to the previous version, bird 3 and bird 4 (also updated Supplementary Fig 4c accordingly).

We also added a very brief pointer to this bird in the main text (page 4, lines 157-158).

6. The pitch oscillations (e.g. in Fig. 3D panel 2) are interesting but appear inconsistent with purely greedy phonology learning. Can the authors please clarify?

We are currently unable to explain these pitch oscillations, but we believe they may arise from conflicts between old and new targets. It may be that birds do not simply give up an old target in favor of a new one at a single point in time. Perhaps, this internal process of target switching has some nontrivial dynamics that would need to be addressed in a separate study.

Reviewer #2 (Remarks to the Author):

This study addresses how the zebra finch, a vocal learning songbird species, copes with the need to learn both a vocabulary of vocal gestures and to organize these gestures into a temporal sequence, in order to match a target model. This important question has relevance for understanding how animal groups acquire their vocal repertoires by imitation, and for speech and language acquisition in humans. The study takes advantage of the detailed knowledge on song structure and learning in zebra finches, and the evidence indicates that these birds can separately correct spectral vs temporal mismatches. They appear to prioritize correcting spectral mismatches of individual syllables over the temporal sequence, when given tasks that in a way forces them to make a choice. The findings provide new insights, the methods are adequate and the evidence seems convincing. While this is a significant contribution, there are some issues with data interpretation, and the authors jump to some conclusions without enough data. These aspects need to be better addressed and some statements toned down.

We thank the reviewer for his/her useful comments, all of which we have addressed, see detailed responses below.

This is a study of zebra finches, not of birds in general, as some passages seem to imply (e.g. “To test which strategy birds employ to learn their song...”). It is unclear whether the findings apply to all vocal learning birds, even less to birds in general. A very large number of bird species are thought to be vocal learners and their songs have very different acoustic structures. Little is known about how these song patterns emerge, thus one should be cautious in assuming that the rules being uncovered here have general applicability. Examining at least another species with a different song structure (e.g. motifs with repeated syllables, different motif structure or syllable sequences, etc) would have helped, but this point is not even discussed.

We have changed the wording throughout the manuscript from “birds” to “zebra finches” or “the birds” as appropriate (substitutions are highlighted throughout the text). It is true that at present the only evidence we obtained is from zebra finches’ song learning, and we now point this out in the discussion (main text, page 10, lines 378-379). We believe that the learning rules we discovered are likely to be applicable to other vocal learning species (see our response to the reviewer’s point on that on page 12 below) and even to other, non-song, complex motor sequences, since these rules stem from a general computational constraint. We have revised the relevant sentence in the discussion (page 10, lines 378-379) to point out that this idea is speculative at this point. The revised sentence (highlighted) reads:

We speculate that zebra finches’ modular vocal learning strategy is an evolutionary compromise between the need to efficiently use motor plasticity resources, and the computational burden of searching through a large space of possible solutions for the most efficient one. We do not know whether the same strategy is employed by other species of vocal learners, or in non-vocal learning processes, but in principle, such trade-off problems are inherent to any case of learning complex motor sequences (e.g.⁶³), and therefore may pose a common constraint shaping the evolution of motor learning mechanisms.

The authors suggest that the rules uncovered might apply to motor learning in general, and provide an analogy with learning to play tennis, but it is unclear whether this is a valid comparison. What would be the equivalent in terms of discrete gestures in this non-vocal motor skill? What is the evidence that such gestures are separate elements that could be learned and/or produced separately? This comparison seems to be quite a stretch

We replaced the tennis-game analogy with an example of learning a new word in a foreign language (main text, page 2, lines 50-53). The revised paragraph reads:

Natural learning of complex behaviors often requires adapting both the structure and the order of gestures in a sequence (Fig. 1a), which is a more complex task than adapting either gestures or sequence alone¹⁰⁻¹⁴. Consider the example of learning a complex word in a foreign language: if one’s utterance is misunderstood, is it because some speech sounds (e.g., phonemes) were pronounced incorrectly (a structural error), or because they were performed in the wrong order (a timing error), or maybe some combination of both? Finding an optimal way to reduce both structural and temporal performance errors constitutes a quadratic assignment problem (Supplementary Text). Such optimization problems are computationally intractable, meaning there is no known efficient algorithm for solving them¹⁵⁻¹⁷.

In contrast, there is no discussion or even mention of what might be the general learning constraints that non-avian vocal learning animals might face in acquiring their vocal repertoires. Because arguably similar auditory-motor constraints are involved, such cases might be of more relevance here than discussing the acquisition of non-vocal motor skills.

We thank the reviewer for this suggestion. We have revised the last section of the discussion (main text, page 10, lines 375-395) to shift the focus from the general problem of motor learning to the problem of learning very complex vocal behaviors. We now discuss the possible role of the learning constraints we discovered in the acquisition of very large (and even infinite) vocal repertoires, such as human language. We believe that a context “deaf” learning module, like the one we discovered, is essential for acquiring such a hugely complex vocal behavior, as it can enable an infant to extract a small set of basic learning targets (e.g. phonemes or syllables) from the highly variable input of her adult tutors, and thus acquire a vocabulary of speech sounds that can be reused for the flexible construction of sequences such as words and sentences.

The revised section (main text, page 10, lines 375-395) replaces the concluding section of the discussion in the previous version. It reads:

A possible role for phonological greediness in learning highly complex vocalizations

We speculate that zebra finches’ modular vocal learning strategy is an evolutionary compromise between the need to efficiently use motor plasticity resources, and the computational burden of searching through a large space of possible solutions for the most efficient one. We do not know whether the same strategy is employed by other species of vocal learners, or in non-vocal learning processes, but in principle, such trade-off problems are inherent to any case of learning complex motor sequences (e.g.⁶³), and therefore may pose a common constraint shaping the evolution of motor learning mechanisms. A structurally greedy workaround, such as we observe in zebra finches, may be particularly useful for the learning of complex behaviors in which combinatorial flexibility is essential. For example, many vocal learners (including songbird species such as nightingales and starlings) have much greater degrees of vocal combinatorial complexity than zebra finches. In particular, combinatorial diversity is a crucial characteristic of human languages, all of which consist of very large word vocabularies, from which an infinite number of higher order sequences can be generated. Thus, human infants’ learning targets are the variable utterances of adults, in which basic speech sounds appear in diverse sequential contexts. Even ignoring the semantic and grammatical aspects of language, the task that human infants face of learning such highly diverse vocal repertoires from scratch seems extremely challenging. A sequence-independent and phonologically greedy vocal learning module may enable infants to extract a small set of basic learning targets (phonemes or syllables) from the highly variable input of adult tutors, and thus facilitate the acquisition of the speech sound vocabulary of their language, presumably occurring at the vocal babbling stage of infant development⁶⁴.

The experimental paradigm of sequential learning of songs that differ by a discrete element is clever, as it allows for dissecting the capabilities and constraints of the vocal learning apparatus of finches. However, it does not exactly reflect what a young zebra finch naturally encounters or the tasks it has to solve. At the start of vocal learning juveniles have not yet acquired their own song, and normally they may not need to modify an acquired song to match a moving target song. So while the study may show what finches are capable of doing, and thus illuminate some features of their learning apparatus, it does not necessarily show the normal pathway that vocal learning follows.

We revised the MS to explain the limitations of our method, also at the requests of Reviewers #1 and #3. It is true that during natural development, when the song is not yet structured, some learning strategies are employed which we did not observe due to our sequential tutoring method, for example the duplication and differentiation of early syllable prototypes (see Reviewer #1’s point 2.1 above). We now address this in the Discussion (main text, page 9, lines 338-346; see also our response to the reviewer’s point about independent modification of syllable renditions in task 3 on page 14 below; to Reviewer #1’s point 2.1; and to Reviewer #3’s point 1). The relevant parts are highlighted:

Our serial tutoring paradigm is confined to mid and late stages of song development, at which the early phenomenon of duplication and differentiation of syllable prototypes to match multiple targets^{22,24,25,29,30} is no longer observed. In our experiments, when a syllable was offered two spectrally similar targets (tasks 3 and 5), only one of those targets was selected - the syllable never “split” to match both. We therefore do not have direct

evidence as to what target assignment algorithm may govern the duplication and differentiation of early proto-syllables, and even on whether proto-syllable splitting is related to target assignments at all or occurs prior to targets being assigned. We cannot rule out the possibility that the “musical chairs” constraints we discovered might not (or not fully) apply to early song development.

We would like to comment, though, that learning a song in a piecewise manner, and from more than one tutor, is not wholly unnatural for zebra finches. Zebra finch predation rate in the wild is very high⁴ and therefore it is probably not uncommon for tutors to be replaced during song development in the wild. In the laboratory, young zebra finches who were switched from the presence of one adult male to another in mid development, acquired the new male’s song either fully or partially⁵. Moreover, it was found that zebra finches raised in an aviary containing 10 breeding pairs (a natural-like setting for zebra finches, who are colonial breeders in the wild) tend to copy and incorporate into their song elements from more than one adult male’s song⁶. These findings suggest that the incomplete imitation of the father’s song that was found in wild zebra finches⁷ may result at least in part from the tendency to copy from other tutors as well. In addition, previous studies found that even birds that are exposed to a song of a single tutor often learn it in a piecewise manner^{8,9}. For example, often not all the syllables of the song are learned at the same time, but some syllables are learned earlier, and others are added later and incorporated into an existing song sequence (a developmental trajectory that requires both structural and temporal adjustments). Given this evidence, we believe that our experimental paradigm of serial tutoring does not require zebra finches to demonstrate capabilities that are not at all employed in their natural vocal development, both with respect to learning a song piece-by-piece and with respect to learning from more than one tutor. We now point this out in the introduction (main text, page 3, lines 86-89). The relevant paragraph reads (revised part is highlighted):

To test which strategy zebra finches employ to learn their song, we used artificial tutoring to generate temporal and spectral mismatches between a bird’s song and its target^{23,26,31,32}. We utilized the fact that under certain conditions (presumably the loss of a tutor due to high predation rates³³) zebra finches can learn their song in a piecewise manner^{25,26} from more than one tutor³⁴⁻³⁶; we trained young male zebra finches with playbacks of an artificially designed tutor song (see methods), and once we could reliably identify copies of all tutored syllables in the singing performance, we switched the training to an altered synthetic song (Fig. 1d). We continuously recorded the birds’ vocal output and tracked the developmental trajectory of individual syllables, to uncover the underlying assignment of performance error and the manner in which it is minimized.

In some experiments, the generality of some conclusions is unclear. In expt. 3, it is interesting that different birds shifted the modified target syllable in different directions. Lowering the pitch of syllable B would have met the demands of both the spectral and temporal matches, however some birds chose an upward shift for this position. Could this be due to a salience effect of the last (in this case 4th) target syllable? Would the same be seen if the syllable that challenged the temporal sequence were at a temporally more neutral position of the motif? This is not to suggest that the authors need to test all possible interchanges, which might require using a longer motif with more syllables, but just to indicate that even with all manipulations provided, only a subset of possible combinations were tested, thus the conclusions should be tempered as well.

We revised the Results (main text, page 5, lines 185-187) to discuss the possibility of a salience effect influencing target selection. As the reviewer points out below, task 4 constitutes a control for the salience hypothesis, since that task includes a source syllable for which the acoustically close out-of-context target is the last syllable in the target motif as well as a syllable for which it is the second syllable in the motif; we did not observe a salience effect in this task, and we now point this out. The revised paragraph reads:

Six out of seven birds shifted pitch towards the acoustically closer (1 semitone) targets (final pitch shift of 0.8 ± 0.2 semitones toward target; Fig. 4b-e; Supplementary Fig. 2). Since the closer targets were at “incorrect” temporal positions, this choice resulted in a reduced match of the target syntax ($-91 \pm 6\%$ relative to source song, task 4.1, Fig. 4d), or in an incomplete match ($58 \pm 7\%$, task 4.2, Fig. 4e). One-semitone targets were chosen regardless of their position in the target motif (2nd or 4th syllable in the motif, Fig 4b-c; Supplementary Fig. 2), indicating that the choice was not a result of a salience effect of the last syllable in the motif. Thus, although we failed to bias the pitch trajectories of song syllables by offering the birds a greater sequence match, even small differences in local acoustic distance affected the choice of targets for phonological error correction. These results confirm the conclusion that zebra finches employ a dedicated mechanism for learning the syllable vocabulary of their target song. Further, this mechanism is acoustically greedy, namely, it prioritizes local acoustic match over global syntax match.

Still regarding this experiment, the example on Fig. 3d, 2nd graph, is intriguing. This bird seems to be able to almost instantaneously switch the pitch of syllable B, suggesting that it has both versions of this syllable in its repertoire, but is trying to find which one would result in the best temporal sequence match.

We now realize that our plotting of daily pitch medians in Fig 3d is visually misleading, and we revised the panel to show median pitch in individual syllable renditions instead (see revised panel below). Although the pitch oscillations in the second bird were fast with respect to the entire developmental period, they were not instantaneous. It took the bird two days and 539 syllable renditions to shift the pitch from B to B⁻, and five days and 1583 syllable renditions to shift the pitch from B⁻ to B⁺. The pitch shifted smoothly, and there was no time in which versions of both target syllables were in the bird's repertoire. This can be seen in the pitch trajectory in Supplementary Fig 3a (Bird 3, each black dot represents a single rendition of syllable B), and in the revised Fig 3d. Eventually syllable B settled on the target B⁻, and the other target B⁺ was matched by an independent precursor (Fig 5e and Supplementary Fig 3a; see also our response to the reviewer's next point below). The updated panel d in Fig 3 now looks:

We agree that the question of the *cause* of the pitch oscillations between the two targets is intriguing, but at present we cannot tell whether the oscillations stem from an attempt to optimize the sequence match, as the reviewer suggests, or from another cause. We suspect that they may arise from conflicts between old and new targets. It may be that birds do not immediately give up an old target in favor of a new one, and that the process of target switching may have some nontrivial dynamics. We are planning to investigate this issue in a separate study (see also Reviewer 1's comment on this above, minor point 6).

While somewhat anecdotal, this seems to illustrate well the point that syllables may be treated as independent elements that can be modified separately, and/or used in multiple combinatorial arrangements. In this experiment, however, it is unclear whether the birds eventually learned or not a motif with 4 syllables, as in the second target song. If not, why not? If a 4th syllable was eventually added, what signal did it have, the correct one? Was this a sequential process?

Here too our presentation of task 3's results was not clear, because we first focused on the choice of target of syllable B, and deferred the issue of how the remaining target was matched to a later section dealing with competitive matching of "vacant" targets (pages 5-6, "Competition among syllables and among targets over syllable-target assignments", Fig 5c-f, Supplementary Fig 3a). We now revised the text accordingly (main text, page 4, lines 152-154). Three out of the 4 birds trained with task 3 learned the missing fourth syllable, using a separate precursor (i.e., *not* syllable B) to match the missing target (but only two of them incorporated it in the correct position in the motif, see Supplementary Fig 4c). We now point this fact earlier on page 4, lines 152-154. The revised text (highlighted) reads:

All four birds shifted the pitch of syllable B, approaching either B⁻ or B⁺, with a remarkable accuracy of 97±1%, defeating the hypothesis of error averaging and indicating a winner-take-all target selection strategy. **The remaining target syllable (towards which syllable B did *not* shift) was matched as well in 3 of the 4 birds, by a precursor initially performed outside of the song motif (see more on this below and in Fig 5 and Supplementary Fig 3).**

We also revised a sentence in the section that deals with matching "vacant" targets to clarify that we are talking about the birds trained with tasks 3 and 5 (p 6, line 215):

We next examined if and how did the birds **in both tasks** match the remaining target (B⁺ or B⁻), which was not selected by syllable B.

The reviewer's point about independent modification of syllable renditions is very relevant to early stages of song development, which we could not observe in this study given our two-stage experimental paradigm. Early on, zebra finches can modify the renditions of a single "proto-syllable" independently to match different targets, but this ability is probably lost in later stages of song development (see Reviewer #1's comment on duplication and differentiation of syllable prototypes on page 4 above). We have revised the results' text and discussion to address this issue (main text, page 4, lines 154-159; page 5, lines 198-200; pages 9-10, lines 337-374; see our responses to the reviewer's point about the limitations of our method above, and to Reviewer #1's points 1 and 2.1 and minor point 5).

The issue of possible high salience of the last syllable is partially addressed in expt 4, since in that case the 4th syllable seems to shift towards an internal (2nd) syllable. Task 4.2 seems less convincing in this regard, since one cannot know for sure if the shift of either B syllable was towards the 2nd or 4th target B syllables, in both cases the direction of the shift would be the same and what matters more is the size of the shift. What is unclear is whether the birds would continue shifting their syllables beyond the period shown here. Did the birds crystallize their songs, or did they show further pitch shifts beyond the period shown?

To make sure that the birds in this group crystallized their song by the end of the experiment, we extended the period of recording way beyond the end of the sensitive period for song learning reported in the literature (which is 90-100 days post hatch⁴); Birds trained with task 4.2 were recorded until day 121, 121, 128, 130 and 153 post hatch (we added this information to the legend of Supplementary Fig 2b, **Supplementary Information**, page 13, lines 365-369). At these ages, males are sexually mature, and perform a crystallized song motif, which remains unchanged for the remainder of their lives (based both on the literature, and on our extensive experience with playback experiments in isolated birds). Therefore, it is highly unlikely that the 4 birds in this group that shifted the B variants towards the acoustically closer 1-semitone targets were on the way to matching the farther 2-semitone targets. Note also that birds trained with other tasks successfully accomplished 2-semitone pitch shifts within the experimental period. We revised the methods (main text, page 11, lines 421-422) and the legend of Supplementary Fig 2b (showing the pitch trajectories of birds in group 4.2) to clarify this point (**Supplementary Information**, page 13, lines 365-369). The revised legend reads:

Supplementary Figure 2

Pitch trajectories in birds trained with tasks 4.1 and 4.2. Median pitch in consecutive renditions of the two pitch shifted syllable types in birds trained with task 4.1 ($ABCB^{+1} \rightarrow AB^{+2}CB^{-1}$) (a) and task 4.2 ($AB^{+2}CB^{-1} \rightarrow ABCB^{+1}$) (b), not including the two birds depicted in Fig 4c. Notation as in Fig 4c. In all birds except one (bird 5 in (b)), the pitch of both syllable types shifted towards the acoustically closer targets. In bird 5, the pitch of both syllable types (shown in black and green for visual clarity) shifted towards the acoustically farther targets. **Bird ages at the end of the experimental period in task 4.2 were 121, 121, 128, 130 and 153 days post hatch. As the sensitive period for song learning in zebra finches ends around day 90-100 post hatch, it is unlikely that birds in this group that matched the 1-semitone targets were on the way to matching the farther targets.**

The relevant sentence in the methods read:

Recording and training were done using Sound Analysis Pro^{38,65}, and continued until birds reached adulthood (day 99-158 post hatch). **At these ages, males are sexually mature, and perform a crystallized song motif, which remains unchanged for the remainder of their lives³³.**

Specific points:

Abstract:

The authors comment: "and searching for the optimal transformation quickly becomes computationally intractable." From the bird's perspective, this seems irrelevant, it solves the problem whether or not a clear computational solution is immediately clear. This issue reflects more a limitation in current methods that a biological problem per se. This statement should probably not be in the Abstract, or at least should be modified.

What we meant to say is that the song learning strategy that evolved in birds is unlikely to optimally trade sequence versus phonology costs. We have changed the sentence to:

However, because both the structure and the temporal position of individual gestures are adjustable, the number of possible motor transformations increases exponentially with sequence length. Identifying the optimal transformation is a computationally intractable problem even for relatively short sequences, raising the question of what strategies have evolved as a workaround.

In: “to correct conflicting phonological and sequential mismatches in song syllables” it will be unclear to the reader that the mismatches are between the already acquired song and a second target model; perhaps the paradigm could be better explained here.

We changed the sentence to:

Here we test how zebra finches cope with the computational complexity of song learning, by prompting juveniles to modify a learned song to correct conflicting phonological and sequential mismatches in song syllables with respect to a newly introduced target song.

The authors speculate: “and could perhaps be a generic solution in the evolution of motor learning mechanisms.” This is quite a stretch, it would be more relevant to speculate that this may be a generic solution to vocal learning mechanisms.

We changed “generic” to “common” to tone the sentence down, but, with the reviewer’s permission, we would like to keep “motor” here. To our knowledge we are the first to have trained animals on a motor learning problem of intractable complexity, and we hope that our findings will motivate similar studies of other complex learned behaviors, both vocal and non-vocal. The workaround we found is extremely elegant (neglecting the quadratic terms of the problem). We speculate this elegant solution will be broadly applicable thanks to its simplicity and robustness

In several places the authors refer to ‘birds’ instead of zebra finches, or even songbirds, they should be more accurate throughout the paper. If referring to the birds in the present study, they should use ‘the birds’ instead of ‘birds’.

Done. See our response to the reviewer’s comment on this above (page 11).

The authors state: “Once birds copied it...”, but it is unclear what was the criteria for learning at the time when the target song was switched. Did the birds need to reach a minimum level of imitation of the first target song? How accurate or variable was the birds’ imitation of that first song? Since there is considerable learning variation at any given age, were all birds at a somewhat uniform stage in the learning? This is important to better understand until what age or stage the birds would be capable of the vocal plasticity being shown here.

We revised the methods to explain our criteria for learning the source song (main text, pages 10-11, lines 411-414). Zebra finches indeed vary considerably in learning rate and success, and so for our serial tutoring method we had to select relatively fast learners of the source song (see methods, page 11, lines 415-419). To assess imitation degree, we measured the percent of similarity³ to the source song model in 10 randomly chosen song bouts on a given day. We considered the source song as being learned when the similarity to the model was at least 70%. At this level of imitation song performance was fairly stable as can be seen in the stack plots in figures 2-5 and Supplementary figures 1 and 3. This revised part of the methods (pages 10-11, lines 411-414) reads:

From day 33-39 until day 43 birds were passively exposed to 20 playbacks per day of the source song, occurring at random with a probability of 0.005 per second. On day 43, each bird was trained to press a key to hear song playbacks, with a daily quota of 20. Once birds learned the source song, we switched to playbacks of the target song. Learning of the source was assessed by quantifying the percent of similarity (Sound Analysis Pro^{38,65}) between the bird’s song motifs and the source model motif in 10 randomly chosen song bouts per day. We considered the source song as being learned when the similarity to the model was at least 70%.

In: “one transition at a time (Fig. 2c)...”, this passage is unclear. The graph indicates that the birds started singing some syllable transitions (BA, CB) not found in the initially learned song well before they started singing

the AC transition, which is necessary for completing the new acquired sequence. This probably means that the birds started singing more varied types of motifs, or incomplete renditions with 2 syllables only, but this is not clear from the text. This point should be more explicitly stated and explained.

This is true. In the course of acquiring a new syntax, zebra finches learn new syllable transitions one by one, with large time lags between the acquisition of each new transition⁹. In the process, they indeed sing somewhat variable sequences, consisting of the new transitions they learned, and some or all of the old (source) transitions, which are necessary to avoid short (2 syllables) song bouts. For example, the bird shown in Fig 2b and c acquired the transition BA early on, so it was sometimes performing A after B, and not C; but since we were interested in the context in which the pitch shift in syllable C occurred, we included in the raster plot in Fig 2b only motifs that included syllable C. We now point this more clearly in the Figure's legend. The relevant part reads:

Figure 2: Acoustic error correction in individual syllables disregards global similarity.

(a) Song models used for imitation task 1 $ABC \rightarrow AC^+B$ (a single motif is shown; birds were trained with two motif repetitions). The pitch of syllable C^+ in the target song is shifted up by two semitones with respect to C in the source. (b) Developmental singing trajectory of an experimental bird trained with imitation task 1. Stack plot shows consecutive renditions of song motifs containing C/C^+ syllables, over experimental days (day 0, switch to target training; instances of the transition BA, which the bird acquired early on, see (c), are excluded from this plot). Colors, pitch of C/C^+ ; grayscale, Wiener entropy in neighboring syllables. Example sonograms of the bird's song at experiment start (bottom) and end (top). The bird first changed its song to ABC^+ (sonogram on the right) and only afterwards corrected the syntax to AC^+B (arrows).

We also clarified this point in the results text (page 3, lines 110-111), which now reads:

All three birds shifted the pitch of syllable C to the target pitch C^+ before making any changes in song syntax. This resulted in a performance of a song they never heard ($A B C \rightarrow A B C^+$; Fig. 2b, Supplementary Audio 2-4), in which the “correct” syllables were sung in the “wrong” order. It took ^{the} birds several additional days to permute syllable order (fully or partially) towards the target syntax $A C^+ B$, one transition at a time (Fig. 2c), incorporating each new transition into their song and performing it in combination with the existing (source) transitions²⁶. We tested an additional nine birds with a slightly more complex task involving pitch mismatches in two different syllables.

In the literature the term ‘acoustic’ is usually meant to refer to both spectral and temporal features. Here the authors seem to be using it as equivalent to the spectral aspect only. This should be clarified, or perhaps switched to spectral, structural, or frequency-related.

We changed the wording from “acoustic” to “spectral” or “phonological”, as appropriate, throughout the text (changes are highlighted).

In: “even when phonological errors are almost as small as production variability”, the authors introduce a concept that had not yet been discussed in the paper, namely that there is measurable variability in vocal production patterns. For this to make sense, some definition of the naturally occurring production variability and how it is measured should be presented, and then compared in terms of amplitude to the measured ‘phonological errors’.

We measured the standard deviation of the median pitch across renditions of the pitch shifted syllables (i.e., syllables A and C in task 2, syllable B in task 3, etc.) on switch day, and found that it ranges between 0.2 and 0.5 semitones across birds, so is actually considerably smaller than 2 semitones. Therefore our statement that “phonological errors [of 2 semitones] are almost as small as production variability” is mistaken, and we removed it from the text. We thank the reviewer for alerting us to this point.

Minor:

We tested AN additional nine birds

Done

$A B C B+1 \leftrightarrow A B+2 C B-1$

This notation in the pdf gives the impression that there is a break in the sequence, the letters should be moved closer together for clarity.

Done

Reviewer #3 (Remarks to the Author):

In this article, Lipkind et al. address the broad question of how animals modify their learned motor sequences to match both sequence information and structural information. They address this question by using the zebra finches as their model system. Specifically, they train juvenile zebra finches to produce a particular sequence of syllables (source) and then once they have learnt this, they switch to a new target that typically involves both a sequence change and a phonology change (pitch shift). Using a few different types of target sequences, they show that zebra finches change the phonology first and this phonology change results in a sequence error that is eventually corrected in a subset of birds. Using this data, they suggest that birds prioritise phonology learning over sequence learning, thereby solving what would have otherwise been a computationally intractable task – learning both phonology and sequence simultaneously. They also suggest that these two might represent two separate modules for learning sequences.

Overall, the paper addresses an important and interesting question. The authors address this question very nicely with appropriate experiments and analysis. Their claims are well supported by the data. Their results are novel and are likely to be of interest to the songbird community and to a wider audience. For the most part, the paper is well written and easy to follow. However, I have a few minor concerns that are listed below.

We thank the reviewer, and have addressed all concerns, see detailed responses below.

1. I am assuming that under natural conditions, zebra finches do not decide to switch to a different song midway through their learning. Therefore, this task could be a little un-natural for them. Clearly young birds cope with this and do learn the new target song and this is definitely a nice way to address the original question of how sequence and structure are both learned. However, whether similar processes are involved in natural learning is not fully discussed by the authors.

This concern was also raised by Reviewers #1 and #2, and we now address it in the Introduction (main text, page 3, lines 86-89) and Discussion (page 9, lines 338-346). Our experimental paradigm indeed prevented us from observing some processes that occur in natural song development, such as the duplication and differentiation of early syllable prototypes (See Reviewer #1's point 2.1 above), and possibly also a "whole motif" learning strategy. We now address these issues in the discussion (pages 9-10, lines 337-374; also see response to the reviewer's next point below). However, a piecewise learning of a song from different tutors is not wholly unnatural for zebra finches. Natural song development often occurs in a piecewise manner, where parts of the song are learned early, and missing syllable or syllables are added later. Learning from more than one tutor also occurs in natural settings in zebra finches, possibly due to loss of tutors given high predation rate in the wild⁴. We now point this out in the introduction (page 3, lines 86-89; see our response to Reviewer #2's point on this on page 13 above).

The authors briefly discuss other literature related to song learning in juveniles in the Results section, but they do not fully discuss possible reasons for the difference in strategy. After all, one prominent strategy for song learning (Liu et al, 2004 - ref. #22) shows that young birds can use a "motif" strategy where they set down a temporal sequence and slowly modify the acoustic structure. The authors could expand on their discussion of reasons for the difference in strategy that they observe.

We have added a section to the discussion (main text, pages 9-10, lines 337-374), in which we discuss how our findings may apply to learning strategies employed during early song development, including the motif learning strategy (page 9, lines 355-363; see also our response to Reviewer 1's points 1.2 and 2.1 above). It is true that our serial tutoring paradigm did not allow us to test learning strategies during very early development, and therefore we cannot reliably extend our conclusions to early phases of development where different learning strategies may be employed. Liu et al observed frequent cases of "motif learning", but another study of

natural song development, Okubo et al 2015, observed motif learning rarely, and showed that it originated in an earlier phase of prototype duplication and differentiation (see R1's point 1.2 above). Rare cases of "motif learning" are consistent with a strategy that prioritizes phonology over syntax, since sometimes minimizing phonological distances could result in a correct syntax by chance. We cannot at present account for the difference between these studies, and think that a conclusive answer to the question of target matching algorithms during early development (which we could not test with our serial tutoring paradigm) would require parallel recording of premotor activity and auditory responses of HVC projection neurons. We now discuss these issues in the paragraph added to the discussion (main text, pages 9-10, lines 337-374), which reads:

A previous study reported variable learning strategies in young zebra finches, including a "motif-learning" strategy²², where syllables in mature zebra finch song develop from very early precursors already arranged in the correct sequential order. This suggests that zebra finches are capable of employing a global alignment target matching strategy (Fig 1c, left) at early stages of their development. Another study of early development observed rare cases of motif learning²⁵, and showed that they originated from an earlier stage of duplication and differentiation of a single proto-syllable precursor, which is consistent with a non-global phonologically greedy strategy (since phonologically greedy target matching could lead to correct sequence matching by chance in rare cases). The challenge in resolving the question of target assignment strategies during early development, and in particular of testing the predictions of our model for early syllable precursors, lies in being able to identify specific assignments between variable proto-syllable renditions and their targets as early as possible. A recent step towards achieving this goal²⁵ was recording the activity of neurons in the premotor song nucleus HVC during singing in young zebra finches, allowing tracking of the process by which a single proto-syllable splits into two or more neurally distinct syllable precursors. Interestingly, HVC premotor neurons have been shown to exhibit precise auditory responses to the target song in young songbirds^{31,62}, possibly providing an instructive signal for song development³¹. Therefore, combining recordings from HVC during singing in juveniles with recordings of auditory responses to the target song in the same neurons may, in the future, allow to track the process by which targets are assigned to early syllable prototypes.

2. They cite an older study (Lipkind et al. 2013) as evidence that syntax learning is not age-dependent, but a large number of birds in that study also did not fully learn syntax changes. It appears like syntax learning capabilities could be limited by age. Did the authors observe any correlation between the day on which playback was switched to the target motif and the extent of syntax learning?

We did not observe a correlation between switch day and learning success ($R=-0.13$; $p=0.58$), and we now mention this in the results (main text, page 7, lines 259-260). However, as learning new syllable transitions is a slow and laborious process (both early and late in development), learning success depends on the number of new transitions that need to be learned. For example, in tasks 4.1 and 4.2, birds needed to acquire 4 new transitions to correct the syntax errors resulting from their greedy matching strategy, and not surprisingly, none of them managed to fully accomplish this task having only half of their normal developmental period available for it. In addition, as we point out in the results (main text, page 4, lines 129-132; and page 7, lines 260-263), the need adjust phonology as well as syntax and birds' tendency to prioritize phonological adjustments, probably also affected syntax learning success and could explain the lower success rate in comparison to Lipkind et al 2013.

3. In this study, the authors use pitch learning as a proxy for learning acoustic structure. Zebra finch syllables have varying degrees of acoustic complexity and I wonder if modifying pitch can be generalized to modifying acoustic structure in general? This is again a potential difference between normal song learning (earlier studies), where birds have to learn more complex acoustic structure (which can sometimes involve pitch changes too, but not the only change). Pitch changes are obviously easier to track and follow from an experimenter's perspective, but do the authors think this would generalize to all kinds of acoustic changes?

It is true. We used pitch to track phonological error correction, and have no direct evidence on adjustment of other spectral features. However, given that the kind of neural architecture that could carry out the phonologically greedy learning strategy that we discovered differs from networks that could underlie alternative learning strategies (e.g., context dependent learning, see "Neural implications" section in the discussion), the possibility that birds have distinct neural apparatuses for learning pitch and learning other spectral features seems unlikely.

4. In the Methods, the authors mention that they switched to playbacks of the target song once the birds learned the source song. Can the authors specify the criteria that they used to assess whether birds had learned the source song or not?

As a criterion for learning the source song we used a similarity threshold of 70% calculated from 10 randomly chosen song bouts on a given day, we now point this out in the methods (main text, pages 10-11, lines 411-414; see also response to R2's comment on this on page 16 above).

5. Finally, in the abstract, the authors state " ... resulting in unnecessary sequence errors that were later corrected". Given that a large number of birds do not fully correct sequence errors and do not achieve perfect imitation of the target sequence, this seems inaccurate to me. It could be changed to " ... unnecessary sequence errors, some of which were later corrected" or something to that effect emphasizing that all errors were not corrected.

Done

1. Vallentin, D., Kosche, G., Lipkind, D. & Long, M. A. Inhibition protects acquired song segments during vocal learning in zebra finches. *Science* (80-.). **351**, 267–271 (2016).
2. Derégnaucourt, S., Mitra, P. P., Fehér, O., Pytte, C. & Tchernichovski, O. How sleep affects the developmental learning of bird song. *Nature* **433**, 710–6 (2005).
3. Tchernichovski, O. & Mitra, P. P. Sound analysis Pro user manual. (2004).
4. Zann, R. A. *The Zebra Finch, A Synthesis of Field and Laboratory Studies*. (Oxford University Press, 1996).
5. Eales, L. Song learning in zebra finches: some effects of song model availability on what is learnt and when. *Anim. Behav.* **33**, 1293–1300 (1985).
6. Williams, H. Models for song learning in the zebra finch: fathers or others? *Anim. Behav.* (1990). doi:10.1016/S0003-3472(05)80386-0
7. Zann, R. Song and call learning in wild zebra finches in south-east Australia. *Anim. Behav.* (1990). doi:10.1016/S0003-3472(05)80982-0
8. Okubo, T. S., Mackevicius, E. L., Payne, H. L., Lynch, G. F. & Fee, M. S. Growth and splitting of neural sequences in songbird vocal development. *Nature* (2015). doi:10.1038/nature15741
9. Lipkind, D. *et al.* Stepwise acquisition of vocal combinatorial capacity in songbirds and human infants. *Nature* **498**, 104–8 (2013).

REVIEWERS' COMMENTS:

Reviewer #1 (Remarks to the Author):

This is a very strong re-submission. All of my major concerns are met. I encourage publication without further delay.

I have only one very minor comment, which the authors can choose to address if they wish.

In section, 'Target matching across timescales' the authors conclude that by positing, without reference, that the syllable may be the discrete unit representing the transition point from phonologically greedy to context dependent target matching. Yet I wonder if the authors would consider the subsyllabic units identified behaviorally as break-points in Strobe light experiments (Cynx, 1990) and, mechanistically, as discrete HVC chains identified with HVC stimulation experiments (Wang et al, 2008). These studies both show that many syllables (especially long ones) can consist of discrete ~50-100 ms duration subunits, which relate to triggering of unique HVC chains (frequently contralateral) and could plausibly represent the transition point discussed above. i.e. within chain learning is context independent, but chains can be rearranged. The prediction would be constraints on learning across song break points (defined for example by strobe light interruptions).

Reviewer #3 (Remarks to the Author):

The authors have very nicely addressed all of my concerns and I recommend the manuscript for publication.

REVIEWERS' COMMENTS:

Reviewer #1 (Remarks to the Author):

This is a very strong re-submission. All of my major concerns are met. I encourage publication without further delay.

We thank the reviewer for the thorough criticism and valuable suggestions, which have greatly improved this manuscript.

I have only one very minor comment, which the authors can choose to address if they wish.

In section, 'Target matching across timescales' the authors conclude that by positing, without reference, that the syllable may be the discrete unit representing the transition point from phonologically greedy to context dependent target matching. Yet I wonder if the authors would consider the subsyllabic units identified behaviorally as break-points in Strobe light experiments (Cynx, 1990) and, mechanistically, as discrete HVC chains identified with HVC stimulation experiments (Wang et al, 2008). These studies both show that many syllables (especially long ones) can consist of discrete ~50-100 ms duration subunits, which relate to triggering of unique HVC chains (frequently contralateral) and could plausibly represent the transition point discussed above. i.e. within chain learning is context independent, but chains can be rearranged. The prediction would be constraints on learning across song break points (defined for example by strobe light interruptions).

We have revised the discussion according to the reviewer's suggestion (main text, page 8, lines 362-367). The relevant paragraph reads (addition highlighted):

What are the smallest units at which spectral learning is independent of temporal context? We find a phonologically greedy and context independent target matching strategy at the time scale of syllable vocabulary, but what happens at time scales within individual syllables? Does context independence continue all the way to the level of the smallest controllable song segments (5-10 ms)⁶¹, or does it break down at a certain point along the song hierarchy? The answer depends on which are the smallest units within the song that a bird is able to rearrange. Namely, units that cannot be "moved around" with respect to each other must be learned in the appropriate context, since in such a case, sequence errors resulting from greedy learning could not be corrected. Given the laborious and time-consuming nature of sequence rearrangements²⁶, learning individual vocal gestures in a context-independent manner would be a highly inefficient strategy, as it would require a large number of positional rearrangements. In addition, resolving competitive conflicts between spectrally similar targets at such small timescales might be difficult, because many more conflicts would need to be considered, thus substantially increasing the number of computations to be performed. A more reasonable strategy would be to switch from context "deaf" to context dependent learning at small timescales. Previous evidence of in-situ differentiation of sub-syllabic elements during zebra finch development²⁴ suggests that the transition point from phonologically greedy to context-dependent target matching may be the song syllable. This would imply that zebra finches could not rearrange the positions of sub-syllabic elements within their song. However, large (50-100 ms long) sub-syllabic notes have been shown to constitute behavioral breaking points⁶², and are thought to be carried out by distinct neural activation chains in the premotor song nucleus HVC⁶³. Therefore an alternative hypothesis is that such sub-syllabic notes are rearrange-able, and that their spectral structure is learned in a phonologically greedy manner. These hypotheses could be tested in the future with appropriate serial tutoring tasks.

Reviewer #3 (Remarks to the Author):

The authors have very nicely addressed all of my concerns and I recommend the manuscript for publication.

We thank the reviewer for the very useful comments that have greatly improved this manuscript.